# Graphically interpreting how incision thresholds influence topographic and scaling properties of modeled landscapes

Nikos Theodoratos[1] and James W. Kirchner[1,2]

[1]Dept. of Environmental Systems Science, ETH Zurich, Zurich, 8092, Switzerland
[2]Swiss Federal Research Institute WSL, Birmensdorf, 8903, Switzerland

*Correspondence to*: Nikos Theodoratos (theodoratos.niko@gmail.com)

**Abstract.**    We examine the influence of incision thresholds on topographic and scaling properties of landscapes that follow a landscape evolution model (LEM) with terms for stream-power incision, linear diffusion, and uniform uplift. Our analysis uses three main tools. First, we examine the graphical behavior of theoretical relationships between curvature and the steepness index (which depends on drainage area and slope). These relationships plot as straight lines for the case of steady-state landscapes that follow the LEM. These lines have slopes and intercepts that provide estimates of landscape characteristic scales. Such lines can be viewed as counterparts of slope–area relationships, which follow power laws in detachment-limited landscapes, but not in landscapes with diffusion. We illustrate the response of these curvature–steepness-index lines to changes in the values of parameters. Second, we define a Péclet number that quantifies the competition between incision and diffusion, while taking the incision threshold into account. We examine how this Péclet number captures the influence of the incision threshold on the degree of landscape dissection. Third, we characterize the influence of the incision threshold using a ratio between it and the steepness index. This ratio is a dimensionless number in the case of the LEM that we use, and reflects the fraction by which the incision rate is reduced due to the incision threshold; in this way, it quantifies the relative influence of the incision threshold across a landscape. These three tools can be used together to graphically illustrate how topography and process competition respond to incision thresholds.

## 1    Introduction

Processes that shape landscapes leave topographic signatures, which can often be visualized by plotting different topographic metrics against one another. An example is the relationship between river gradient and drainage area, which has been used to analyze landscapes and river profiles, and to diagnose the processes that shape them (e.g., Montgomery and Foufoula-Georgiou, 1993; Howard, 1994; Montgomery and Dietrich, 1994; Dietrich et al., 2003). For example, the stream-power incision model predicts that if tectonics, climate, and rock properties are uniform, then bedrock rivers should approach a steady state in which their gradient scales as a power law of drainage area (e.g., Tucker, 2004; Lague, 2014). This power-law scaling implies that river gradient data should plot as a straight line against drainage area data on logarithmic axes. The properties of this line can give estimates of properties of the landscape, e.g., its slope gives the concavity index (Whipple, 2004). Plotting synthetic topographic data from landscape evolution models (LEMs) helps to illustrate the effects of different model formulations or parameterizations. For example, including a threshold in the incision term of an LEM affects the resulting slope–area line (e.g., Tucker, 2004; Lague et al., 2005; Deal et al., 2018).

In the case of landscapes that are influenced by diffusion, topographic slope does not scale as a power function of drainage area (e.g., Howard, 1994). Thus, slope and area data from these landscapes do not plot as straight lines. In Theodoratos et al. (2018), we presented a counterpart relationship for the case of landscapes produced by an LEM that includes linear diffusion (along with stream-power incision and uplift). This relationship predicts that in steady state, curvature and the steepness index (which depends on drainage area and slope; e.g., Whipple, 2001) plot as a straight line against each other on linear (i.e., non-logarithmic) axes. The slope and intercept of this line depend on characteristic scales of length and height of the landscape, which in turn depend on the relative strengths of the processes that shape it. Thus, this relationship predicts a link between topographic and scaling properties of landscapes that follow the LEM.

Here, we demonstrate an example of the explanatory power of plots of the curvature–steepness-index relationship. Our example shows that these plots can visualize topographic and scaling effects of incision thresholds. Incision thresholds can markedly influence erosion, as shown by numerous studies. For instance, incision thresholds can influence the relationship between river gradient and the uplift rate (e.g., Snyder et al., 2003), the dependence of long-term erosion rates on the average, the variability, and the duration of precipitation events (e.g., DiBiase and Whipple, 2011; Scherler et al., 2017), and the dynamics of migrating knickpoints (e.g., Lague, 2014). Here, we are not further elaborating on the insights of these studies. Instead, we focus on the effects of incision thresholds on the competition between incision and diffusion, and on the topographic and scaling properties of landscapes reflecting this competition. The topographic and scaling effects that we examine have been studied before (e.g., Montgomery and Dietrich, 1992; Howard, 1994; Tucker, 2004; Perron et al., 2008). Here, however, we present a novel, purely graphical method to identify, quantify, and interpret these effects based on the relationship between curvature and the steepness index.

In Theodoratos et al. (2018), we dimensionally analyzed a frequently used LEM with terms for uplift, linear diffusion, and stream-power incision without an incision threshold. In Theodoratos and Kirchner (2020), we added an incision threshold to this LEM and dimensionally analyzed it. Here, we summarize the definitions of characteristic scales and dimensionless numbers that emerged from the dimensional analyses of these two LEMs in Sect. 2. Then, in Sect. 3, we show that these characteristic scales and dimensionless numbers have geomorphologic meaning that can be expressed graphically using plots of curvature versus the steepness index. The graphical explanatory power of these plots is further highlighted by comparing plots of LEMs with and without an incision threshold (Figs. 1 and 2).

## 2 Stream-power incision and linear diffusion LEMs

### 2.1 Governing equations

The LEM without incision threshold follows the governing equation (e.g., Howard, 1994; Dietrich et al., 2003):

$$\frac{\partial z}{\partial t} = -K\sqrt{A}|\nabla z| + D\nabla^2 z + U \quad . \tag{1}$$

This equation gives the rate of elevation change $\partial z/\partial t$ as the sum of three terms, namely, a) stream-power incision $K\sqrt{A}|\nabla z|$, where $K$ is the incision coefficient, $A$ is drainage area, and $|\nabla z|$ is topographic slope, b) linear diffusion $D\nabla^2 z$, where $D$ is the diffusion coefficient and $\nabla^2 z$ is the Laplacian curvature, and c) the uplift rate $U$. We assume that Eq. (1) has base dimensions of horizontal length L, height H (which we treat as dimensionally distinct from L), and time T. All quantities in Eq. (1) have dimensions that are combinations of L, H, and/or T, which we show in Table 1.

Note that the incision term $K\sqrt{A}|\nabla z|$ is a special case of the more general incision term $KA^m(|\nabla z|)^n$. As we explained in Theodoratos et al. (2018), dimensional analysis of an LEM with generic exponents $m$ and $n$ would lead to equivalent results as the analysis of Eq. (1), but these results would be expressed with much more complicated mathematical formulas.

Therefore, in Theodoratos et al. (2018) we focused on the case of exponents $m = 0.5$ and $n = 1$ and we presented the results for generic $m$ and $n$ in an appendix. Likewise, in the current study, the main presentation focuses on the case of $m = 0.5$ and $n = 1$, and in Appendix A we demonstrate that our graphical method is also valid for the case of generic exponents $m$ and $n$.

Following Perron et al. (2008), we can add an incision threshold to the LEM by recasting the incision term as

$K(\sqrt{A}|\nabla z| - \theta)$, where $\theta$ is the incision threshold. This formulation assumes that the incision rate $K\sqrt{A}|\nabla z|$ is reduced everywhere by the constant quantity $K\theta$. The LEM examined here is based on the assumption that sediment transport is detachment limited. Thus, it does not include deposition and negative incision rates would not be meaningful. Therefore, the incision term is set to zero where the term $K(\sqrt{A}|\nabla z| - \theta)$ would be negative, i.e., where $\sqrt{A}|\nabla z| \le \theta$, and the governing equation becomes

$$\frac{\partial z}{\partial t} = \begin{cases} D\nabla^2 z + U \quad, & \sqrt{A}|\nabla z| \le \theta \\ -K(\sqrt{A}|\nabla z| - \theta) + D\nabla^2 z + U \quad, & \sqrt{A}|\nabla z| > \theta \end{cases}. \tag{2}$$

The incision threshold $\theta$ has the same dimensions as $\sqrt{A}|\nabla z|$, i.e., dimensions of H.

Equation (2) assumes that precipitation rates are constant in time and uniform in space, and it incorporates climatic effects into the incision coefficient $K$. Other LEMs use stochastic precipitation to drive their incision terms (e.g., Tucker, 2004, Whipple, 2004; Lague et al., 2005; DiBiase and Whipple, 2011; Deal et al., 2018). The incision thresholds of these LEMs

define limiting values of shear stress or stream power, below which no incision occurs. At any given location in the landscape, these limiting values might be exceeded during some stochastic events and not exceeded during other events, depending on their intensities. By contrast, in the case of the LEM that we examine, the assumption of constant and uniform precipitation implies that any given combination of drainage area $A$ and slope $|\nabla z|$ would lead to the same value of stream power (or shear stress) for any storm event (as all events would be equal), and this value of stream power would either be

above or below the incision threshold. In this idealized case, defining a topographic threshold based on $\sqrt{A}|\nabla z|$ is exactly equivalent to defining a threshold of stream power (or shear stress).

We acknowledge that the LEMs with stochastic precipitation allow much more realistic integration of incision rates over time, compared to the LEM that we examine here. Therefore, these LEMs are more appropriate for studying the influence of

incision thresholds on erosion rates compared to the LEM that we use. However, our study has a different focus. Our study focuses on how the incision threshold $\theta$ influences topographic and scaling properties of landscapes, and on how this influence can be graphically expressed with curvature–steepness-index lines. For these tasks, the simplified formulation of the incision term of Eq. (2) is more practical. It may be possible in future work to extend this approach to include incision thresholds driven by stochastic precipitation.

## 2.2 Characteristic scales

The two governing equations (Eqs. 1 and 2) can be non-dimensionalized using characteristic scales of length, height, and time $l_c$, $h_c$, and $t_c$, defined as (Theodoratos et al., 2018; Theodoratos and Kirchner, 2020)

$$l_c := \sqrt{D/K} \quad , \tag{3}$$
$$h_c := U/K \quad , \tag{4}$$
$$t_c := 1/K \quad . \tag{5}$$

We summarize these and other definitions of this presentation in Table 2. The characteristic scales $l_c$, $h_c$, and $t_c$ can be viewed as intrinsic properties of a landscape, in the sense that they depend exclusively on the values of the parameters $K$, $D$, and $U$, and not on extensive properties of the landscape such as the size of its domain or its maximum relief. We present geomorphologic interpretations of these characteristic scales in Sect. 3.

By combining $l_c$, $h_c$, and $t_c$ we can define additional characteristic scales (Theodoratos et al., 2018). For example, given that drainage areas $A$ have dimensions $L^2$, we can define a characteristic area $A_c$ as the square of the characteristic length:

$$A_c := l_c^2 = D/K \quad . \tag{6}$$

Likewise, we can define a characteristic gradient $G_c$

$$G_c := h_c/l_c = U/\sqrt{DK} \quad , \tag{7}$$

and a characteristic curvature $\kappa_c$

$$\kappa_c := h_c/l_c^2 = U/D \quad . \tag{8}$$

## 2.3 Incision-threshold number $N_\theta$

In Theodoratos and Kirchner (2020), we derived a dimensionless number, whose definition and interpretation we summarize here. Dimensional analysis of the governing equation with incision threshold $\theta$ (Eq. 2) yielded the dimensionless grouping of parameters $K\theta/U$. Specifically, all terms of Eq. (2) give rates of elevation change and have dimensions of H T$^{-1}$. Therefore, to non-dimensionalize Eq. (2), we divided all of its terms by the uplift rate $U$. The quantity $K\theta$, which is included in the incision term of Eq. (2) and gives the reduction in the rate of incision due to the threshold, also has dimensions of H T$^{-1}$. Therefore, dividing the incision term of Eq. (2) by $U$ yielded the dimensionless ratio $K\theta/U$. We defined this dimensionless ratio as an incision-threshold number $N_\theta$

$$N_\theta := K\theta/U \quad . \tag{9}$$

This analysis led to a dimensionless version of Eq. (2) that includes only one parameter, the incision-threshold number $N_\theta$. This implies that $N_\theta$ is a control on the topography of landscapes that follow Eq. (2). Specifically, model landscapes that have equal incision-threshold numbers $N_\theta$ can be set up such that they follow geometrically similar evolutions. Model landscapes that have different $N_\theta$ cannot evolve geometrically similarly, and their topographies differ in ways that depend on their $N_\theta$ values. Simulation results illustrating these points are presented in Theodoratos and Kirchner (2020).

We proposed two interpretations of the incision-threshold number $N_\theta$ in Theodoratos and Kirchner (2020). First, $N_\theta$ is defined as the incision rate reduction $K\theta$ relative to the uplift rate $U$ (Eq. 9). The uplift rate $U$ can be viewed as a characteristic rate of elevation change because it is equal to the ratio of the characteristic height to the characteristic time, i.e., $U = h_c/t_c$ (Eqs. 4, 5). Consequently, $N_\theta$ is a normalized incision rate reduction with respect to $U$. Second, if we rearrange Eq. (9) as $N_\theta = \theta/(U/K)$, then we can interpret $N_\theta$ as giving the magnitude of $\theta$ relative to the parameter ratio

$U/K$. Thus, the definition of $N_\theta$ shows that incision thresholds from different landscapes should not be compared to each other according to their own values, but instead according to their values relative to the ratio $U/K$ of each landscape.

## 3  Graphical illustrations of topographic and scaling effects of the incision threshold

### 3.1  Defining a steady-state topographic relationship between the steepness index and curvature

In Theodoratos et al. (2018), we presented a relationship that describes the steady-state topography of landscapes that evolve according to Eq. (1). Specifically, if we set $\partial z/\partial t = 0$ and we solve the governing equation for curvature $\nabla^2 z$, we obtain

$$\partial z/\partial t = 0: \qquad \nabla^2 z = (K/D)\sqrt{A}|\nabla z| - (U/D) \quad . \tag{10}$$

The quantity $\sqrt{A}|\nabla z|$ is equal to the steepness index (defined as $A^{m/n}|\nabla z|$ for drainage area and slope exponents $m$ and $n$; e.g., Whipple, 2001). For this reason, we refer to Eq. (10) as the curvature–steepness-index relationship.

In a coordinate system in which the steepness index ($\sqrt{A}|\nabla z|$) and curvature ($\nabla^2 z$) are plotted on the horizontal and vertical axes, respectively, Eq. (10) plots as a straight line (for example, see Fig. 1, which we describe in more detail further below). Equation (10) is a testable, quantitative prediction; if a landscape is in steady state and has evolved according to Eq. (1), then curvature should plot as a straight line against the steepness index. Furthermore, this line can give estimates of the parameters $K$, $D$, and $U$, because its slope is $K/D$, and its intercepts are $\nabla^2 z = -U/D$ and $\sqrt{A}|\nabla z| = U/K$. While we have

not validated this prediction with data, Eq. (10) is a rearranged version of Eq. (5) in Perron et al. (2009), which has been successfully tested with real-world landscape data and has been used to estimate model parameters. Testing Eq. (10), and Eqs. (11–12) and Figs. 1–2, which are described further below, would be a reasonable next step after the current study.

### 3.2  Characteristic scales and the curvature–steepness-index relationship

If we substitute the characteristic scales $l_c$ and $\kappa_c$ for the parameter ratios $K/D$ and $U/D$, then the curvature–steepness-index
relationship (Eq. 10) becomes

$$\partial z/\partial t = 0: \qquad \nabla^2 z = (1/l_c^2)\,\sqrt{A}|\nabla z| - \kappa_c \quad . \tag{11}$$

As this equation shows, an interpretation of $l_c$ and $h_c$ is that they control steady-state topography. Specifically, for a landscape to be in steady state, drainage area $A$, topographic slope $|\nabla z|$, and curvature $\nabla^2 z$ must obey Eq. (11), which is parameterized by the characteristic scales $l_c$ and $\kappa_c$, or equivalently by $l_c$ and $h_c$ because $\kappa_c = h_c/l_c^2$ (Eq. 8). We can graphically illustrate the control of $l_c$ and $h_c$ on the topography by plotting the curvature–steepness-index line described by
Eq. (11). As Fig. 1 shows, the properties of such a line are controlled by $l_c$ and $h_c$, specifically, its slope is $1/l_c^2$, and its intercepts are $\nabla^2 z = -\kappa_c = -h_c / l_c^2$ and $\sqrt{A}|\nabla z| = h_c$. Note that the slope of this line can be represented either as $K/D = 1/l_c^2$ units of curvature per 1 unit of steepness index, or 1 unit of curvature per $D/K = l_c^2$ units of steepness index. For simplicity, we use the latter notation to express the slopes of the curvature–steepness-index lines in Figs. 2–4.

Likewise, the curvature–steepness-index relationship that corresponds to the LEM with incision threshold $\theta$ is controlled by the characteristic scales $l_c$ and $\kappa_c$. This relationship, however, is also controlled by the incision-threshold number $N_\theta$. To derive this relationship, we set $\partial z/\partial t = 0$ in Eq. (2) and we solve it for $\nabla^2 z$. When we do this for the second subdomain

(where $\sqrt{A}|\nabla z| > \theta$), we encounter the ratio $K\theta/D$. This ratio can be rewritten as $K\theta/D = (K\theta\, U)/(U\, D) = N_\theta\, \kappa_c$ (Eqs. 8, 9). Thus, we obtain the curvature–steepness-index relationship:

$$\partial z/\partial t = 0: \quad \begin{cases} \nabla^2 z = -\kappa_c \ , & \sqrt{A}|\nabla z| \leq \theta \\ \nabla^2 z = (1/l_c^2)\,\sqrt{A}|\nabla z| - (1 + N_\theta)\kappa_c \ , & \sqrt{A}|\nabla z| > \theta \end{cases} . \qquad (12)$$

We plot this equation in Fig. 2 in black and, for comparison, we also plot the curvature–steepness-index line without incision threshold (Eq. 11) in gray. The black line consists of two segments that correspond to the two subdomains of Eqs. (2) and (12). The first segment is horizontal and describes a uniform steady-state curvature value of $\nabla^2 z = -\kappa_c$ for points with $\sqrt{A}|\nabla z| \leq \theta$, where incision is fully suppressed by the threshold and only diffusion and uplift operate. The second segment is inclined and corresponds to points with $\sqrt{A}|\nabla z| > \theta$ where all three processes operate.

Equations (11–12) and Figs. 1–2 show that the characteristic scales $l_c$, $h_c$, and $\kappa_c$ describe the steady-state topography at points of special interest (see also Theodoratos et al., 2018). Furthermore, some effects of incision thresholds on landscape properties can be visualized by comparing the curvature–steepness-index lines with and without an incision threshold (black and gray lines of Fig. 2).

First, the vertical-axis intercept of the curvature–steepness-index line without incision threshold (Fig. 1, Eq. 11) corresponds to ridges and drainage divides, which have $A = 0$ and/or $|\nabla z| = 0$, i.e., $\sqrt{A}|\nabla z| = 0$. This intercept shows that the steady-state curvature of ridges and drainage divides is $\nabla^2 z = -\kappa_c = -U/D$ (see also Roering et al., 2007; Perron et al., 2009). Note that $-\kappa_c$ is the most negative value of curvature. The horizontal segment of the black line in Fig. 2 (described by the first subdomain of Eq. 12) expresses the fact that, in landscapes with an incision threshold $\theta$, the points with $\sqrt{A}|\nabla z| \leq \theta$ have the same steady-state curvature as ridges and drainage divides, i.e., the most negative value of curvature. This shows that adding an incision threshold to the LEM results in more convex hillslopes (e.g., Howard, 1994; Theodoratos and Kirchner, 2020).

Second, the curvature–steepness-index line without incision threshold (Fig. 1, Eq. 11) has a horizontal-axis intercept of $\sqrt{A}|\nabla z| = h_c$. This intercept corresponds to points with curvature $\nabla^2 z = 0$, which can be viewed as defining the transition between hillslopes and valleys (e.g., Howard, 1994). Thus, points with steepness index equal to the characteristic height $h_c$ can be used to map hillslope–valley transitions (Theodoratos et al., 2018). Adding an incision threshold $\theta$ to the LEM makes landscapes steeper and decreases the drainage density, i.e., makes first-order basins bigger, (e.g., Montgomery and Dietrich, 1992; Howard, 1994; Perron et al., 2008). These two effects lead to steeper gradients $|\nabla z|$ and larger drainage areas $A$ at hillslope–valley transitions. Specifically, as Fig. 2 shows, the horizontal-axis intercept increases from $\sqrt{A}|\nabla z| = h_c$ (gray line) to $\sqrt{A}|\nabla z| = h_c + \theta$ (black line).

### 3.3  Quantifying and visualizing the effect of the incision threshold on the scales of landscape dissection

In Theodoratos et al. (2018), we derived an interpretation of the characteristic length $l_c$ by analyzing the competition between the advection and diffusion of elevation perturbations (e.g., knickpoints), which gives rise to ridges and valleys, and controls their characteristic sizes (e.g., Smith and Bretherton, 1972; Howard, 1994; Perron et al., 2008). Following Perron et al. (2008, 2009, 2012), we quantified the relative strength of advection versus diffusion using a Péclet number Pe. The

definition of our Péclet number differs somewhat from Perron et al.'s. Specifically, our definition includes a length scale $l$ that we termed flow path length and that we defined as the distance along flow paths from a given point to the farthest ridge.

The Péclet number is defined (e.g., Perron et al., 2008) as the ratio of a diffusion timescale $t_D$ to an incision timescale $t_I$, each of which gives a measure of the strength of the respective process. Specifically, a diffusion timescale can be defined as (e.g., Perron et al., 2008)

$$t_D := \frac{l^2}{D} \ . \tag{13}$$

This timescale characterizes diffusive propagation over a distance $l$. In Theodoratos et al. (2018), to define $t_I$, we first calculated the celerity $c$ that corresponds to the incision term of Eq. (1), which is a kinematic wave term (e.g., Whipple and Tucker, 1999). This celerity is equal to $c = K\sqrt{A}$. Perturbations can be assumed to be advected at this celerity (e.g., Berlin and Anderson, 2007; Perron et al., 2008). Lague (2014) has criticized this assumption because it does not take into account the effects of knickpoints on hydraulics (e.g., on stream width) and their feedbacks on the rate of knickpoint propagation, especially in the presence of incision thresholds. While we acknowledge this limitation, we nonetheless assume that the rate of knickpoint advection is equal to the celerity $c$ of Eq. (17) because our current focus is on interpreting the characteristic scales $l_c$, $h_c$, and $t_c$, which pertain to Eqs. (1) and (2), which do not describe hydraulics explicitly. Therefore, in Theodoratos et al. (2018), we defined the incision timescale $t_I$ as the ratio of the flow path length $l$, which characterizes the location of points within drainage basins, to the celerity $c$, which characterizes the strength of advection:

$$t_I := l/c = \frac{l}{K\sqrt{A}} \ . \tag{14}$$

Note that small values of $t_I$ and $t_D$ correspond to strong advection and diffusion, respectively.

We can quantify the relative strengths of advection and diffusion using the ratio of the respective timescales, which defines the Péclet number (Theodoratos et al., 2018):

$$Pe := t_D/t_I = \frac{\sqrt{A}\, l}{l_c^2} = \frac{\sqrt{A}}{\sqrt{A_c}} \frac{l}{l_c} \ . \tag{15}$$

Diffusive propagation is stronger at points with Péclet number smaller than 1 and advective propagation is stronger where the Péclet number is larger than 1. Where the Péclet number is roughly equal to one, diffusion and advection will be roughly equal (when measured by $t_D$ and $t_I$). Equation (15) shows that if a point's flow path length $l$ is roughly equal to the characteristic length $l_c$ and its drainage area $A$ is roughly equal to the characteristic area $A_c$, then its Péclet number will be roughly equal to one. i.e.,

$$l \approx l_c, \qquad A \approx A_c \approx l_c^2 \qquad \Longrightarrow \qquad Pe \approx 1 \ . \tag{16}$$

Note that if the incision term has a slope exponent $n \neq 1$, then the condition $|\nabla z| \approx G_c$ must be included along with $l \approx l_c$ and $A \approx l_c^2$ for the Péclet number to be $Pe \approx 1$.

The conditions $A \approx l_c^2$ and $l \approx l_c$ (Eq. 16) are not the only combination of $A$ and $l$ that give $Pe \approx 1$, but they are significant because they lead to an interpretation of $l_c$. Specifically, these conditions show that advective propagation, which promotes valley dissection, is dominant at points farther than $l_c$ from the ridge and with drainage area greater than $l_c^2$. Therefore, in Theodoratos et al. (2018) we interpreted the characteristic length $l_c$ as giving a measure of the smallest scales of dissection. This interpretation does not imply that valley heads are exactly 1 $l_c$ away from ridges or that they have drainage areas exactly

equal to 1 $l_c^2$. Rather, it implies that flow path lengths and drainage areas of valley heads are of similar order of magnitude as $l_c$ and $l_c^2$, respectively. Furthermore, it implies that valley heads in different landscapes have $l$ and $A$ that scale with $l_c$ and $l_c^2$, respectively.

Adding the threshold $\theta$ to the incision term of the LEM changes the kinematic wave celerity $c$ and, thus, the incision timescale $t_I$ and the Péclet number Pe. Specifically, the celerity becomes

$$c = \begin{cases} 0 \ , & \sqrt{A}|\nabla z| \leq \theta \\ K\sqrt{A} - K\theta/|\nabla z| \ , & \sqrt{A}|\nabla z| > \theta \end{cases} , \qquad (17)$$

and, thus, the incision timescale $t_I$ becomes

$$t_I := l/c = \begin{cases} +\infty \ , & \sqrt{A}|\nabla z| \leq \theta \\ \dfrac{l}{K\sqrt{A} - K\theta/|\nabla z|} \ , & \sqrt{A}|\nabla z| > \theta \end{cases} . \qquad (18)$$

Note that the diffusion timescale $t_D$ is not affected by the incision threshold. Thus, we can use Eqs. (13) and (18) to define a Péclet number Pe for the LEM with incision threshold $\theta$ (Eq. 2), specifically

$$\text{Pe} := t_D/t_I = \begin{cases} 0 \ , & \sqrt{A}|\nabla z| \leq \theta \\ \dfrac{\sqrt{A}\,l - (\theta\,l/|\nabla z|)}{l_c^2} \ , & \sqrt{A}|\nabla z| > \theta \end{cases} . \qquad (19)$$

It can be shown that Eq. (19) can be rewritten as

$$\text{Pe} := t_D/t_I = \begin{cases} 0 \ , & \sqrt{A}|\nabla z| \leq \theta \\ \dfrac{\sqrt{A}}{\sqrt{A_c}}\dfrac{l}{l_c} - N_\theta \dfrac{l}{l_c}\dfrac{G_c}{|\nabla z|} \ , & \sqrt{A}|\nabla z| > \theta \end{cases} , \qquad (20)$$

where $N_\theta$ is the incision-threshold number (Eq. 9). Equation (20) shows that adding an incision threshold $\theta$ to the LEM reduces the Péclet number relative to the Péclet number for the LEM without a threshold (Eq. 15). This agrees with the fact that the threshold weakens the incision term. More specifically, the Péclet number for the LEM with $\theta$ is reduced by the quantity $N_\theta(l/l_c)(G_c/|\nabla z|)$.

Note that the Péclet number definition by Perron et al. (2008) also includes a reduction that depends on $N_\theta$ (denoted as $\theta'$ in Perron et al., 2008). The two definitions differ in that ours includes the product $\sqrt{A}\,l$ (where $A$ is the drainage area and $l$ is the flow path length), whereas Perron et al.'s definition includes only a length scale (squared). By including $\sqrt{A}\,l$, our definition can account for the scaling of $A$ with $l$, which depends on the convergence or divergence of topography. The

implications of this property of our Péclet number are discussed in Sect. 4.2.3 of Theodoratos et al. (2018).

Using Eq. (20) we see that the conditions $l \approx l_c$, $A \approx l_c^2$, and $|\nabla z| \approx G_c$, which lead to a Péclet number roughly equal to 1 for the case without incision threshold (Eq. 16), will lead to Pe $\approx 1 - N_\theta < 1$ when $\theta$ is included. The fact that the difference between Pe and 1 is equal to $N_\theta$ suggests that we could obtain the value Pe $\approx 1$ by adjusting the values of $l$, $A$, and $|\nabla z|$ such

that they depend on $N_\theta$. Indeed, we observe that

$$l \approx \sqrt{1 + N_\theta}\,l_c, \qquad A \approx (1 + N_\theta)\,l_c^2, \qquad |\nabla z| \approx \sqrt{1 + N_\theta}\,G_c \quad \Rightarrow \quad \text{Pe} \approx 1 \ . \qquad (21)$$

Note that $\sqrt{1 + N_\theta}\,l_c$ is larger than $l_c$, which agrees with observations that incision thresholds reduce landscape dissection (e.g., Montgomery and Dietrich, 1992; Howard, 1994; Perron et al., 2008).

Equation (21) shows that, in the case of a landscape that includes an incision threshold $\theta$, the smallest scales of dissection are not characterized by the characteristic length $l_c$ on its own, but rather jointly by $l_c$ and the incision-threshold number $N_\theta$ through the quantity $\sqrt{1 + N_\theta}\, l_c$. Consequently, the presence of $\theta$ changes the dependence of the scales of dissection on the

LEM parameters. Without an incision threshold, the scales of landscape dissection depend on $l_c$, which depends on the incision and diffusion coefficients $K$ and $D$ (Eq. 3). On the other hand, when $\theta$ is included in the LEM, the scales of dissection depend on $\sqrt{1 + N_\theta}\, l_c$, which depends on $K$ and $D$, but also on the uplift rate $U$ and the incision threshold $\theta$. We illustrate an example of the dependence on $U$ in Fig. 4 b.

The length scales $l_c$ and $\sqrt{1 + N_\theta}\, l_c$ can be expressed graphically by the horizontal- and vertical-axis intercepts of curvature–steepness-index lines, specifically, by the ratio of these intercepts (or, more precisely, by the ratio of their absolute values). This ratio is equal to $h_c/\kappa_c = l_c^2$ in the case without incision threshold (see Fig. 1) and equal to $(h_c + \theta)/\kappa_c = (1 + N_\theta)l_c^2$ in the case that includes the incision threshold $\theta$ (see Fig. 2). Note that the first ratio is equal to the inverse of the slope of the curvature–steepness-index line, which is $1/l_c^2$. On the other hand, the second ratio is not the inverse of this slope, which

remains $1/l_c^2$ when the threshold $\theta$ is included. Instead, it is the inverse of the slope of an auxiliary line connecting the two intercepts. In Fig. 2, we show this auxiliary line with a black dashed line style. The effect of the incision threshold on valley dissection can be visualized graphically by comparing the slope of the curvature–steepness-index line against the slope of the black dashed auxiliary line. We denote this comparison as a thick white arrow.

It should be noted that the characteristic length $l_c$ depends only on $K$ and $D$ only when the slope exponent is $n = 1$. However, for other values of $n$, $l_c$ will also depend on the uplift rate $U$; in this more general case, $l_c = (K^{-1}D^n U^{1-n})^{1/(n+2m)}$ (see Appendix A in Theodoratos et al., 2018). Therefore, the degree of landscape dissection in general is not independent of $U$. Specifically, an increase of $U$ leads to a decrease of landscape dissection for $n > 1$ and to an increase of landscape dissection for $n < 1$, which agrees with previous observations of the dependence of drainage density on the uplift rate (e.g., Clubb et

al., 2016). Interestingly, as revealed by the current study, if an incision threshold is included, the degree of landscape dissection depends on $U$ even for $n = 1$.

### 3.4 How the curvature–steepness-index line responds to parameter value changes

As we show in Figs. 3 and 4, differences in the properties of landscapes with different parameters $K$, $D$, $U$, and $\theta$ can be graphically summarized by curvature–steepness-index lines, because the slopes and intercepts of these lines depend on the

characteristic scales $l_c$, $h_c$, and $\kappa_c$, and on the incision-threshold number $N_\theta$, which in turn depend on the parameters.

Figure 3 shows curvature–steepness-index lines without incision thresholds. It consists of three panels, each showing how the lines respond to an increase in one of the three parameters $U$, $K$, and $D$. In panel (a), an increase in the uplift rate $U$ shifts the curvature–steepness-index line downward and to the right without changing its slope. This illustrates that the

characteristic height and curvature $h_c$ and $\kappa_c$, which control the intercepts of the line, are proportional to $U$ (Eqs. 4, 8), while the characteristic length $l_c$, which controls the line's slope, is independent of $U$ (Eq. 3). The parallel shift of the line corresponds to more convex ridges (so that diffusion can keep up with uplift), to steeper gradients (so that incision can keep up with uplift), and to unchanged landscape dissection. Analogously, panel (b) shows that an increase in the incision

coefficient $K$ leads to a counterclockwise rotation of the line around the vertical-axis intercept, which corresponds to a more dissected landscape (smaller $l_c$), milder gradients (smaller $h_c$), and unchanged ridge convexity (unchanged $\kappa_c$). Finally, in panel (c), an increase in the diffusion coefficient $D$ results in a clockwise rotation of the line around the horizontal-axis intercept. This corresponds to a smoother landscape with less dissection (larger $l_c$) and less convex ridges (smaller $\kappa_c$), and to

unchanged steepness index at hillslope–valley transitions (unchanged $h_c$).

Figure 4 illustrates in four panels how curvature–steepness-index lines respond to increases in the value of either the incision threshold $\theta$ or one of the parameters $U$, $K$, and $D$. It is reminded that a curvature–steepness-index line with incision threshold consists of two segments, a horizontal and an inclined. Note that, as we explain in the previous subsection (Sect. 3.3), a

curvature–steepness-index line that includes an incision threshold does not express landscape dissection through the slope of its inclined segment, which depends only on the characteristic length $l_c$, but rather through the ratio of the horizontal- and vertical-axis intercepts, which is equal to $\sqrt{1 + N_\theta}\, l_c$. This ratio can be graphically illustrated by the slope of an auxiliary line that connects the two intercepts, such as the dashed black line in Fig. 2. In each panel of Fig. 4, we show two dashed black auxiliary lines to illustrate how the ratio of intercepts responds to the parameter changes.

In panel (a) of Fig. 4, we illustrate an increase in $\theta$. The steepness index $\sqrt{A}|\nabla z|$ must reach a greater value before exceeding the increased $\theta$ and, thus, the horizontal segment of the curvature–steepness-index line becomes longer. The vertical position of this segment (along with the vertical-axis intercept) do not change, because the characteristic curvature $\kappa_c$ does not depend on $\theta$. The slope of the curvature–steepness-index line also does not change, because $l_c$ does not depend on $\theta$. Thus, the

increase of $\theta$ parallel-shifts the inclined segment of the line to the right. Consequently, the horizontal-axis intercept increases, which expresses the steepening of gradients and the decrease of landscape dissection. The decrease of dissection is also expressed by the fact that the ratio of the horizontal- to the vertical-axis intercept increases, as shown by the clockwise rotation of the dashed auxiliary line.

In panel (b) of Fig. 4, we show that an increase in the uplift rate $U$ parallel-shifts the curvature–steepness-index line downward and to the right. Furthermore, the horizontal- and vertical-axis intercepts move to the right and downward, respectively ($\kappa_c$ and $h_c$ are proportional to $U$), and the slope of the inclined segment remains unchanged ($l_c$ does not depend on $U$). As we explain in Sect. 3.3, the value of $U$ affects the value of $\sqrt{1 + N_\theta}\, l_c$, which expresses the scales of landscape dissection. Specifically, the increase of $U$ leads to a decrease of $N_\theta$. This reflects the fact that $\theta$ becomes less important

relative to the increased $U$. Thus, the decrease of dissection due to the threshold is somewhat moderated by the increase of $U$. This moderation is graphically illustrated by the slopes of auxiliary lines connecting the intercepts of the curvature–steepness-index lines. These auxiliary lines do not intersect and, thus, their slopes cannot be readily compared visually. Therefore, we plot them again in an inset such that they start from the same point. In this way, we can see that the increase of $U$ leads to a counterclockwise rotation of the auxiliary lines, which expresses the increase of dissection.


In panel (c) of Fig. 4, we illustrate the response of the curvature–steepness-index line to an increase in the incision coefficient $K$. The horizontal segment of the line remains unchanged and the inclined segment is rotated counterclockwise around the point of transition between the two segments. Likewise, the dashed auxiliary line connecting the horizontal- and vertical-axis intercepts is rotated counterclockwise. These responses express that dissection is decreased and that gradients become milder when $K$ is increased. Finally, in panel (d), we show that increasing the diffusion coefficient $D$ leads to a


clockwise rotation of the inclined segment of the line around its horizontal-axis intercept, which remains unchanged. The rotation results in moving the horizontal segment up and in rotating the dashed auxiliary line clockwise. These changes express the reduction in landscape dissection and the reduction in the convexity of ridges and hillslopes.

## 4 Quantifying how the influence of the incision threshold varies within a landscape

Thus far, we have examined how the influence of the incision threshold $\theta$ varies between different landscapes with different parameters using the incision-threshold number $N_\theta$ (Eq. 9). This number is constant for any given landscape if the parameters of the landscape are constant. Now, we turn our attention to how the influence of the incision threshold $\theta$ varies within a given landscape.

We can quantify the relative influence of the threshold $\theta$ on the rate of incision using the fraction $\theta/(\sqrt{A}|\nabla z|)$. This fraction is equal to $K\theta$, the reduction in the incision rate due to the threshold, divided by $K\sqrt{A}|\nabla z|$, the incision rate if there would be no threshold. Therefore, $\theta/(\sqrt{A}|\nabla z|)$ shows by what fraction the incision rate is reduced due to the threshold. Where $\sqrt{A}|\nabla z| = \theta$, the fraction $\theta/(\sqrt{A}|\nabla z|)$ is equal to 1, which agrees with the incision rate being reduced by 100% (i.e., being reduced to zero). At points with $\sqrt{A}|\nabla z|$ smaller than $\theta$, calculating the fraction $\theta/(\sqrt{A}|\nabla z|)$ would not be meaningful; instead, because the threshold completely suppresses incision under these conditions, we assign a value of 1 to the fractional reduction in incision rate.

We can associate the fractional reduction in incision rate to Tucker's (2004) threshold factor $\Phi$. Tucker (2004) defined $\Phi$ to quantify the fraction of precipitation events leading to shear stress above a threshold value, i.e., the fraction of events that lead to erosion. Tucker (2004) used $\Phi$ to express the incision term of his LEM as $KA^{m_b}S^{n_b}\Phi$. In the case of the LEM examined here (Eq. 2), following Tucker's notation, we can express the incision term as $K\sqrt{A}|\nabla z|\Phi$, where the threshold factor $\Phi$ is equal to $1 - \theta/(\sqrt{A}|\nabla z|)$ for $\sqrt{A}|\nabla z| > \theta$ and to 0 for $\sqrt{A}|\nabla z| \leq \theta$. Thus, the quantity $1 - \Phi$ is equal to the fractional reduction in incision rate, i.e.,

$$1 - \Phi = \begin{cases} 1 \, , & \sqrt{A}|\nabla z| \leq \theta \\ \dfrac{\theta}{\sqrt{A}|\nabla z|} \, , & \sqrt{A}|\nabla z| > \theta \end{cases} . \tag{22}$$

Consequently, in what follows we denote the fractional reduction in incision rate as $1 - \Phi$. We illustrate the properties of the quantity $1 - \Phi$ with plots and maps in Figs. 5–7.

In Fig. 5 we plot $1 - \Phi$ versus the steepness index $\sqrt{A}|\nabla z|$ according to Eq. (22). The curve consists of two parts. The first is a horizontal segment that describes the value $1 - \Phi = 1$ and corresponds to points with $\sqrt{A}|\nabla z| \leq \theta$, where incision is fully suppressed by the threshold. The second part corresponds to points with $\sqrt{A}|\nabla z| > \theta$, forming part of a hyperbola that asymptotically approaches 0. This asymptotic approach expresses the fact that, at points with steepness index $\sqrt{A}|\nabla z|$ much larger than $\theta$, the incision threshold has a very small relative influence on the incision rate.

To indicate how different parts of the $1 - \Phi$ curve of Fig. 5 correspond to different regimes of a landscape, we identify the point that corresponds to hillslope–valley transitions. As explained in Sect. 3.2, hillslope–valley transitions can be defined as points with zero curvature and, therefore, with a steady-state steepness index of $\sqrt{A}|\nabla z| = h_c + \theta$. Consequently, the fractional reduction in incision rates $\theta/(\sqrt{A}|\nabla z|)$ at these points is $\theta/(h_c + \theta)$. We can rewrite this value in terms of the

incision-threshold number $N_\theta$ as $N_\theta/(1 + N_\theta)$. Thus, in Fig. 5, hillslope–valley transitions correspond to the point with coordinates $\left(\sqrt{A}|\nabla z|, 1 - \Phi\right) = \left(h_c + \theta, \; N_\theta/(1 + N_\theta)\right)$, which we mark with a black dot. The part of the curve above and to the left of this dot corresponds to hillslopes, and the part below and to the right corresponds to the valley network.

With Fig. 6, we examine how the value of the incision-threshold number $N_\theta$ of a landscape controls the relationship between

the quantity $1 - \Phi$ and the steepness index $\sqrt{A}|\nabla z|$. Specifically, in Fig. 6 we show curves of $1 - \Phi$ versus $\sqrt{A}|\nabla z|$ for four landscapes with incision-threshold numbers $N_\theta$ equal to 0.2, 0.4, 1, and 2. The landscapes are assumed to have equal parameters $K$, $D$, and $U$, and therefore to have equal characteristic scales. The curves with greater values of $N_\theta$ also have greater incision thresholds $\theta$ and, thus, they have longer horizontal segments. Furthermore, the curves with greater values of $N_\theta$ go towards zero more slowly. On each curve, we show the hillslope–valley transition using a black dot. The value of the

quantity $1 - \Phi$ corresponding to each dot becomes larger as $N_\theta$ increases. Thus, in landscapes with smaller $N_\theta$, the incision rate is reduced by large fractions only on the hillslopes, and in valleys it is reduced by small fractions. By contrast, in landscapes with greater $N_\theta$, incision can be reduced by large fractions both on hillslopes and in valleys.

Figure 7 shows maps of the quantity $1 - \Phi$ across four steady-state landscapes. We simulated these landscapes with the

CHILD numerical model (Channel-Hillslope Integrated Landscape Development model; Tucker et al., 2001). Details about these simulations and additional results are presented in Theodoratos and Kirchner (2020). Here, we provide brief information about the parameters and setup of these simulations in Appendix B. To illustrate how the spatial distribution of $1 - \Phi$ depends on the incision-threshold number $N_\theta$, we ran four simulations with $N_\theta$ values of 0.2, 0.4, 1, and 2, i.e., the same $N_\theta$ values as in Fig. 6. The pixels of the four maps are colored according to their values of $1 - \Phi$ using a grayscale that

ranges from white to black. Lighter colors correspond to larger values of $1 - \Phi$, i.e., to stronger influence of the incision threshold. As expected, lighter colors appear near ridges and on hillslopes, where the incision threshold has a stronger influence.

The patterns in Fig. 7 reflect the spatial distribution of drainage area and slope, because the incision threshold in Eq. (2) is

defined as a topographic threshold. However, maps of the quantity $1 - \Phi$ would be useful for other formulations of the incision threshold, as well. For example, Tucker's (2004) formulation of the incision threshold assumed stochastic precipitation. Tucker quantified the influence of this incision threshold using the threshold factor $\Phi$, which ranges between 0 and 1 (and on which our quantity $1 - \Phi$ is based, as mentioned above). Therefore, the quantity $1 - \Phi$ could be calculated for the case of Tucker's (2004) LEM, and maps of this quantity would visualize how the influence of the incision threshold is

spatially distributed across landscapes.

The fractional reduction in incision rate as $1 - \Phi$ and the threshold factor $\Phi$ can be used to simplify the definition of the Péclet number Pe. Specifically, we can rearrange the definition of Pe (Eq. 19) such that it includes the fraction $\theta/(\sqrt{A}|\nabla z|)$:

$$\text{Pe} = \begin{cases} 0 \quad , & \sqrt{A}|\nabla z| \leq \theta \\ \left(1 - \dfrac{\theta}{\sqrt{A}|\nabla z|}\right) \dfrac{\sqrt{A}}{\sqrt{A_c}} \dfrac{l}{l_c} \quad , & \sqrt{A}|\nabla z| > \theta \end{cases} . \tag{23}$$

If this equation is combined with the definition of $1 - \Phi$ (Eq. 22), then we can rewrite the definition of Pe in compact form

$$\text{Pe} = \Phi \cdot \text{Pe}_{\theta=0} \quad , \tag{24}$$

where $\text{Pe}_{\theta=0}$ is the Péclet number for the LEM without incision threshold (see Eq. 15). Equations (23) and (24) reveal that the influence of the incision threshold on the Péclet number varies across the landscape. Specifically, larger values of Pe, which correspond to larger values of the steepness index $\sqrt{A}|\nabla z|$, are less sensitive to the incision threshold.

## 5    Summary and conclusions

We present graphical methods that summarize topographic and scaling properties of landscapes following a simple stream-power incision and linear diffusion LEM (Eq. 1), and that illustrate the effects of adding an incision threshold $\theta$ (Eq. 2). Our results referring to the LEM without incision threshold (Eq. 1) have been presented before (Theodoratos et al., 2018), but we show them here again to contrast them against those referring to the LEM with the threshold $\theta$ (Eq. 2). The two LEMs (Eq. 1, 2) assume that the incision term has drainage area and slope exponents $m = 0.5$ and $n = 1$, because this combination significantly simplifies the mathematical derivations. However, as we show in Appendix A, our results are also valid for generic exponents $m$ and $n$.

For the first graphical method, we plot steady-state relationships between curvature $\nabla^2 z$ and the steepness index $\sqrt{A}|\nabla z|$ (Eqs. 10, 11, 12), which we obtain from the governing equations Eq. (1) and (2). These relationships can be viewed as counterparts of the relationship between topographic slope and drainage area, which is typically assumed to follow a power law in detachment-limited landscapes, but not in landscapes that are also influenced by hillslope diffusion. These relationships plot as straight lines (Figs. 1 and 2), whose properties (slope and intercepts) depend on the incision threshold $\theta$ and on the characteristic scales of the landscape, which in turn depend on the parameters of the LEM, i.e., on the incision coefficient $K$, the diffusion coefficient $D$, and the uplift rate $U$. (Eqs. 3, 4, and 8). A reasonable follow-up study would be to validate these results against real-world landscapes, and specifically to explore whether curvature and steepness-index data from real landscapes would plot against each other as straight lines.

With Fig. 2, we show that curvature–steepness-index lines can graphically illustrate effects of incision thresholds on landscapes. Specifically, the ways in which curvature–steepness-index lines with and without threshold differ from each other illustrate that thresholds make hillslopes more convex and gradients steeper, and reduce the drainage density. These effects have been presented elsewhere (e.g., Montgomery and Dietrich, 1992; Howard, 1994; Perron et al., 2008), but the curvature–steepness-index lines offer new ways to visualize them graphically. In Figs. 3 and 4, we illustrate the dependence of these properties on the parameters $K$, $D$, $U$, and $\theta$ by showing how curvature–steepness-index lines respond to increases in these parameters, one at a time. These figures demonstrate an advantage of curvature–steepness-index lines: the topographic effects of model parameter changes are expressed as simple shifts and rotations of these lines.

In Sect. 3.3, we examine in more detail the effects of the incision threshold $\theta$ on drainage density and the scales of landscape dissection, and how these effects can be visualized with curvature–steepness-index lines. We assume that dissection is controlled by the competition between the advection and diffusion of perturbations (e.g., Smith and Bretherton, 1972; Howard, 1994; Perron et al., 2008) and, thus, we examine the effects of $\theta$ using a Péclet number Pe (Eqs. 15, 19; see also
Perron et al., 2008, 2012; Theodoratos et al., 2018). For the LEM that does not include an incision threshold, we found in Theodoratos et al. (2018) that the characteristic length $l_c$ characterizes the smallest scales of dissection. Note that the slope of curvature–steepness-index lines is $1/l_c^2$; therefore, this slope graphically expresses the scales of dissection of landscapes without incision thresholds. Adding the incision threshold $\theta$, we find that the smallest scales of dissection are characterized by the length scale $\sqrt{1 + N_\theta}\, l_c$, where $N_\theta$ is a dimensionless incision-threshold number defined as $N_\theta = K\theta/U$ (Eq. 9). This
length scale is longer than $l_c$, which expresses the fact that incision thresholds reduce the drainage density. The square of this length scale is $(1 + N_\theta)\, l_c^2$ and is equal to the ratio between the horizontal- and vertical-axis intercepts of the curvature–steepness-index line. As we show in Fig. 2, an auxiliary line connecting these two intercepts would have a slope of $1/\big((1 + N_\theta)\, l_c^2\big)$. Thus, we can graphically visualize the effect of the incision threshold on landscape dissection by comparing the slope of this auxiliary line with the slope of the curvature–steepness-index line.

The second graphical method consists of plots of the dimensionless fraction $\theta/\sqrt{A}\,|\nabla z|$, which gives the fraction by which the incision rate is reduced due to the threshold (see the governing equation Eq. 2). We found that this fraction is equal to $1 - \Phi$ (Eq. 22), where $\Phi$ is a threshold factor (see Tucker, 2004) that subsumes the effect of the incision threshold $\theta$ on the incision term of the LEM (see Eqs. 2, 22). Thus, we denote the fractional reduction in the incision rate as $1 - \Phi$. In Figs. 5–
7, we present plots and maps of $1 - \Phi$ that illustrate how the relative influence of incision thresholds will vary across a given landscape, and how the variation of this relative influence depends on the landscape's incision-threshold number $N_\theta$.

The two dimensionless numbers examined here, $N_\theta$ and $1 - \Phi$, quantify the relative influence of the incision threshold $\theta$, the first with respect to the parameters $K$ and $U$, and the second with respect to the steepness index. Thus, $N_\theta$ quantifies how $\theta$
affects different landscapes with different parameters, and $1 - \Phi$ quantifies how the influence of $\theta$ varies across different points of a given landscape. We find that the definition of the Péclet number Pe can be rewritten in two equivalent forms (Eqs. 20, 24), which reveal how Pe depends on $N_\theta$ and on $\Phi$, respectively.

The three dimensionless numbers, Pe, $N_\theta$, and $\Phi$, along with the characteristic scales $l_c$, $h_c$, and $t_c$, provide a thorough
characterization of landscapes that follow the governing equation Eq. (2). Furthermore, plots of the curvature–steepness-index relationship offer a straightforward way to graphically express the geomorphologic meaning of these dimensionless numbers and characteristic scales. Even though the specific definitions of these quantities refer only to the LEMs examined here (Eqs. 1, 2), the approach that underpins our graphical methods is more generally applicable. For example, an LEM with incision threshold and stochastic precipitation would have a different governing equation than Eq. (2) and, thus, a different
curvature–steepness-index relationship than Eq. (12) and Fig. 2. However, curvature and the steepness index would still be reasonable axes for plotting data from such an LEM if it included diffusion. Likewise, the quantity $1 - \Phi$ would follow a different formula than Eq. (22), but maps of this quantity would be useful in visualizing spatial patterns of the influence of the incision threshold across a landscape. Consequently, our graphical methods could potentially be helpful for the analysis of a broader range of models than those examined here.

## Appendix A: Curvature–steepness-index lines for generic drainage area and slope exponents $m$ and $n$

In this appendix, we demonstrate that our graphical method remains valid for the case of LEMs with incision terms that have generic drainage area and slope exponents $m$ and $n$.

For generic exponents $m$ and $n$, the governing equations Eq. (1) and Eq. (2) become, respectively,

$$\frac{\partial z}{\partial t} = -KA^m(|\nabla z|)^n + D\nabla^2 z + U \quad , \tag{A1}$$

and

$$\frac{\partial z}{\partial t} = \begin{cases} D\nabla^2 z + U \quad , & A^m(|\nabla z|)^n \leq \theta \\ -K(A^m(|\nabla z|)^n - \theta) + D\nabla^2 z + U \quad , & A^m(|\nabla z|)^n > \theta \end{cases} \quad . \tag{A2}$$

Given that the steepness index is defined as $k_s = A^{m/n}|\nabla z|$ (e.g., Whipple, 2001), the quantity $A^m(|\nabla z|)^n$ in the above equations is equal to the steepness index raised to the power $n$, i.e., $A^m(|\nabla z|)^n = k_s^n$, and the incision threshold $\theta$ is a threshold of the quantity $k_s^n$.

Setting $\partial z/\partial t = 0$ in Eqs. (A1) and (A2), we can derive the corresponding steady-state relationships between curvature and the steepness index:

$$\nabla^2 z = (K/D)\, k_s^n - (U/D) \quad , \tag{A3}$$

and

$$\begin{cases} \nabla^2 z = -(U/D) \quad , & k_s^n \leq \theta \\ \nabla^2 z = (K/D)k_s^n - (1 + N_\theta)\,(U/D) \quad , & k_s^n > \theta \end{cases} \quad , \tag{A4}$$

where $N_\theta$ is the incision-threshold number, still defined as $N_\theta = K\theta/U$.

When plotted in axes of $\nabla^2 z$ and $k_s^n$, Eq. (A3) has the same basic properties as Eq. (11), the curvature–steepness-index relationship for $m = 0.5$ and $n = 1$, and $\theta = 0$. Specifically, Eq. (A3) plots as a straight line with a slope equal to $K/D$, a vertical-axis intercept equal to $-U/D$ and a horizontal-axis intercept equal to $U/K$. Consequently, for generic exponents $m$ and $n$, changes in the values of the parameters $K$, $D$, and $U$ are still expressed graphically as shifts and rotations of the curvature–steepness-index line, as seen in Fig. 3 for the case of $m = 0.5$ and $n = 1$.

Note that the characteristic scales of length and height $l_c$ and $h_c$ are not equal to $\sqrt{D/K}$ and $U/K$ for generic exponents $m$ and $n$. Rather, they are defined by the more complicated formulas:

$$l_c = (K^{-1}D^n U^{1-n})^{1/(n+2m)} \quad , \tag{A5}$$

and

$$h_c = (K^{-2}D^{n-2m}U^{2-n+2m})^{1/(n+2m)} \quad , \tag{A6}$$

(whose derivation can be seen in Appendix A of Theodoratos et al., 2018). However, the parameter ratios $K/D$ and $U/K$ still express the relative strengths of incision versus diffusion, and incision versus uplift. By contrast, note that the parameter ratio $U/D$ remains equal to the characteristic curvature $\kappa_c$, which expresses the relative strength of diffusion versus uplift. Consequently, the shifts and rotations of the curvature–steepness-index line still express changes in scaling and topographic

properties of landscapes, such as changes in curvature of ridges, in degree of dissection, and in gradients at hillslope–valley transitions.

Likewise, when plotted in axes of $\nabla^2 z$ and $k_s^n$, Eq. (A4) has the same properties as Eq. (12), the curvature–steepness-index relationship with incision threshold and with $m = 0.5$ and $n = 1$. Specifically, Eq. (A4) plots as a line with two segments, a horizontal segment at $\nabla^2 z = -U/D$ for $k_s^n \leq \theta$, and an inclined segment with slope equal to $K/D$ and horizontal-axis intercept equal to $(U/K) + \theta$ for $k_s^n > \theta$. This line, too, responds to changes in the parameters $\theta$, $K$, $D$, and $U$ with shifts and rotations, equivalent to the shifts and rotations shown in Fig. 4 for the case of $m = 0.5$ and $n = 1$.

Finally, for generic exponents $m$ and $n$, the fractional reduction in incision rate $1 - \Phi$ is defined as

$$1 - \Phi = \begin{cases} 1 & , \quad k_s^n \leq \theta \\ \theta/k_s^n & , \quad k_s^n > \theta \end{cases} , \tag{A7}$$

which plots as shown in Figs. 5 and 6, but in axes of $k_s^n$.

## Appendix B:  Setup of numerical simulations

We prepared the maps of Fig. 7 with results from numerical simulations that we performed using the CHILD model, originally for the work discussed in Theodoratos and Kirchner (2020). In that work, we present much more information
about these simulations and their results. Here, we briefly summarize the model setup and parameterization.

All four landscapes in Fig. 7 have incision coefficient $K = 2 \times 10^{-6} \text{ a}^{-1}$, diffusion coefficient $D = 0.5 \times 10^{-2} \text{ m}^2\text{a}^{-1}$, and uplift rate $U = 0.5 \times 10^{-4} \text{ m a}^{-1}$. Each landscape's incision threshold $\theta$ depends on the value of its incision-threshold number $N_\theta$ according to $\theta = N_\theta \cdot (U/K)$ (see Eq. 9), where $U/K = 25$ m for all landscapes. Therefore, the landscapes have
the incision thresholds seen in Table B1.

We simulated the four landscapes on triangular irregular networks (TINs) with total extent of 7.5 km × 11.25 km and average TIN edge length of 20 m, which resulted in around a quarter million TIN points. Each map in the left column of Fig. 7, shows a part of the TIN, specifically, a rectangular region with size of 5 km × 4 km, centered around the largest
drainage basin of the corresponding landscape.

Details about the implementation of the governing equation (Eq. 2) in CHILD (Tucker et al., 2001) can be found in Theodoratos et al. (2018) and in Theodoratos and Kirchner (2020).

Table B1: Incision-threshold numbers $N_\theta$ and corresponding incision thresholds $\theta$ of the four landscapes illustrated in Fig. 7. The parameter ratio $U/K$ is equal to 25 m for all landscapes.

| Incision-threshold number: | $N_\theta$ (−) | 0.2 | 0.4 | 1 | 2 |
|---|---|---|---|---|---|
| Incision threshold: | $\theta = N_\theta \cdot (U/K)$ (m) | 5 | 10 | 25 | 50 |

## Data availability

The data presented here were synthesized using the CHILD model (Tucker et al., 2001). The input files needed to reproduce them are available from the corresponding author upon request.

## Author contribution

NT derived and analyzed the theoretical, numerical, and graphical results, and NT and JWK interpreted them. NT drafted the paper, and NT and JWK edited it.

## Competing interests

The authors declare that they have no conflict of interest.

## Acknowledgments

This study was made possible by financial support from ETH Zurich. We thank Eric Deal for helpful discussions.

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

## Tables

Table 1: Descriptions and dimensions of the terms, variables, and parameters in the governing equations Eq. (1) and Eq. (2). Dimensions are expressed in terms of the model's fundamental dimensions of horizontal length L, vertical length (height) H, and time T.

| Symbol | Description | Dimensions |
|---|---|---|
| $\partial z / \partial t$ | Total rate of elevation change at a point $(x, y)$ | $H\,T^{-1}$ |

Rates of elevation change due to:

| Symbol | Description | Dimensions |
|---|---|---|
| $-K\sqrt{A}|\nabla z|$ | stream-power incision (in Eq. 1) | $H\,T^{-1}$ |
| $-K(\sqrt{A}|\nabla z| - \theta)$ | threshold-limited stream-power incision (in Eq. 2) | $H\,T^{-1}$ |
| $D\nabla^2 z$ | linear diffusion | $H\,T^{-1}$ |
| $U$ | uplift | $H\,T^{-1}$ |
| $(x, y)$ | Horizontal coordinates | L |
| $z$ | Elevation | H |
| $t$ | Time | T |
| $A$ | Drainage area | $L^2$ |
| $|\nabla z|$ | Topographic slope | $H\,L^{-1}$ |
| $\nabla^2 z$ | Laplacian curvature | $H\,L^{-2}$ |
| $K$ | Incision coefficient | $T^{-1}$ |
| $D$ | Diffusion coefficient | $L^2\,T^{-1}$ |
| $U$ | Uplift rate | $H\,T^{-1}$ |
| $\theta$ | Incision threshold | H |
| $\sqrt{A}|\nabla z|$ | Steepness index | H |

Table 2: Summary of definitions and formulas used in this study.

| Description | Definition | Equation |
|---|---|---|
| Characteristic length | $l_c := \sqrt{D/K}$ | (3) |
| Characteristic height | $h_c := U/K$ | (4) |
| Characteristic time | $t_c := 1/K$ | (5) |
| Characteristic area | $A_c := l_c^2 = D/K$ | (6) |
| Characteristic gradient | $G_c := h_c/l_c = U/\sqrt{DK}$ | (7) |
| Characteristic curvature | $\kappa_c := h_c/l_c^2 = U/D$ | (8) |
| Incision-threshold number | $N_\theta := K\theta/U$ | (9) |
| Curvature–steepness-index relationship, without $\theta$ | $\nabla^2 z = (K/D)\sqrt{A}|\nabla z| - (U/D)$ | (10) |
| | $\nabla^2 z = (1/l_c^2)\sqrt{A}|\nabla z| - \kappa_c$ | (11) |
| Curvature–steepness-index relationship, with $\theta$ | $\begin{cases} \nabla^2 z = -\kappa_c , & \sqrt{A}|\nabla z| \le \theta \\ \nabla^2 z = (1/l_c^2)\sqrt{A}|\nabla z| - (1+N_\theta)\kappa_c , & \sqrt{A}|\nabla z| > \theta \end{cases}$ | (12) |
| Threshold factor | $\Phi = \begin{cases} 0 , & \sqrt{A}|\nabla z| \le \theta \\ 1 - \dfrac{\theta}{\sqrt{A}|\nabla z|} , & \sqrt{A}|\nabla z| > \theta \end{cases}$ | (22) |
| Fraction of incision rate reduction | $1 - \Phi = \begin{cases} 1 , & \sqrt{A}|\nabla z| \le \theta \\ \dfrac{\theta}{\sqrt{A}|\nabla z|} , & \sqrt{A}|\nabla z| > \theta \end{cases}$ | (22) |
| Flow path length | $l$: the distance along flow paths from a point to the farthest ridge | N/A |
| Diffusion timescale | $t_D := \dfrac{l^2}{D}$ | (13) |
| Kinematic wave celerity, without $\theta$ | $c = K\sqrt{A}$ | N/A |
| Incision timescale, without $\theta$ | $t_I := l/c = l/(K\sqrt{A})$ | (14) |
| Péclet number, without $\theta$ | $\mathrm{Pe}_{\theta=0} := t_D/t_I = \dfrac{\sqrt{A}\, l}{l_c^2} = \dfrac{\sqrt{A}}{\sqrt{A_c}}\dfrac{l}{l_c}$ | (15) |
| Kinematic wave celerity, with $\theta$ | $c = \begin{cases} 0 , & \sqrt{A}|\nabla z| \le \theta \\ K\sqrt{A} - K\theta/|\nabla z| , & \sqrt{A}|\nabla z| > \theta \end{cases}$ | (17) |
| Incision timescale, with $\theta$ | $t_I := l/c = \begin{cases} +\infty , & \sqrt{A}|\nabla z| \le \theta \\ l/(K\sqrt{A} - K\theta/|\nabla z|) , & \sqrt{A}|\nabla z| > \theta \end{cases}$ | (18) |
| Péclet number, with $\theta$ | $\mathrm{Pe} := \begin{cases} 0 , & \sqrt{A}|\nabla z| \le \theta \\ \dfrac{\sqrt{A}}{\sqrt{A_c}}\dfrac{l}{l_c} - N_\theta \dfrac{l}{l_c}\dfrac{G_c}{|\nabla z|} , & \sqrt{A}|\nabla z| > \theta \end{cases}$ | (20) |
| | $\mathrm{Pe} = \begin{cases} 0 , & \sqrt{A}|\nabla z| \le \theta \\ \left(1 - \dfrac{\theta}{\sqrt{A}|\nabla z|}\right)\dfrac{\sqrt{A}\, l}{l_c^2} , & \sqrt{A}|\nabla z| > \theta \end{cases}$ | (23) |

**Figures**

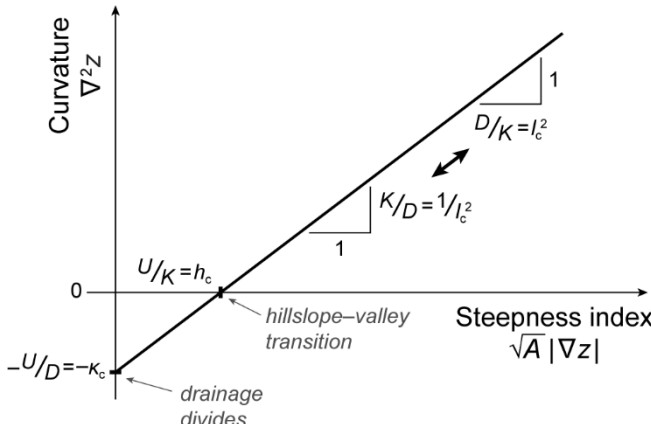

Figure 1: **Relationship between curvature and the steepness index in steady-state topography without incision threshold.** We plot a straight line defined by Eqs. (10) or (11), which describes how curvature should be related to the steepness index if the landscape follows the LEM (Eq. 1) and is in steady state. This line is parameterized by the characteristic scales of length and height $l_c$ and $h_c$ (Eqs. 3, 4); its slope is $1/l_c^2$, its horizontal-axis intercept is $\sqrt{A}|\nabla z| = h_c$, and its vertical-axis intercept is $\nabla^2 z = -\kappa_c$ (where $\kappa_c$ is a characteristic curvature defined as $h_c/l_c^2$; Eq. 8). These intercepts reveal topographic properties of special points in a landscape, namely, the steady-state curvature of drainage divides and the steady-state steepness index of hillslope–valley transitions. The characteristic length $l_c$ quantifies the competition between knickpoint advection and diffusion and predicts how landscape dissection scales with the parameters. Thus, the slope of the curvature–steepness-index line expresses visually how dissected a landscape is. Note that the line's slope can be represented either as $K/D = 1/l_c^2$ units of curvature per 1 unit of steepness index, or 1 unit of curvature per $D/K = l_c^2$ units of steepness index. For simplicity, we use the latter notation to express the slopes of the lines in Figs. 2–4.

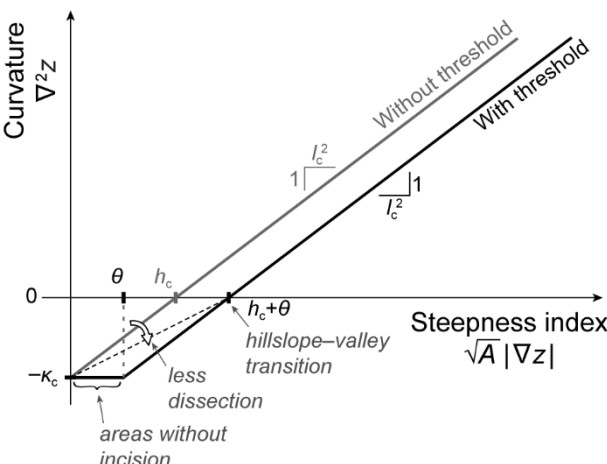

Figure 2: **Effects of incision threshold on steady-state topography as reflected in the curvature–steepness-index line.** We show curvature–steepness-index lines of landscapes with and without incision threshold using black and gray colors, respectively (see Eqs. 11, 12). The gray line in this figure is identical to the line in Fig. 1. Adding an incision threshold to the LEM changes the resulting steady-state topography, as indicated by the differences between the gray and black lines. The black line consists of two segments. The horizontal segment corresponds to points where incision is fully suppressed by the threshold. This horizontal segment is at $\nabla^2 z = -\kappa_c$, the vertical-axis intercept of the gray line. This shows that the hilltop curvature (the most negative curvature value in the landscape) spreads to points on hillslopes beyond drainage divides. Thus, hillslopes become more convex due to the threshold. The inclined segment of the black line is parallel to the gray line, and at a horizontal distance $\theta$ to its right. Thus, the horizontal-axis intercept is increased from $h_c$ to $h_c + \theta$ due to the threshold, i.e., the hillslope–valley transition occurs at a larger steepness index value. This increase corresponds to larger drainage area $A$ and/or steeper slope $|\nabla z|$, both of which are consistent with the steepening of landscapes and the decrease of their drainage density by the incision threshold. In the case of the LEM that includes an incision threshold, the degree of landscape dissection is expressed by the length scale $\sqrt{1 + N_\theta}\, l_c$ (see Eq. 21), where $N_\theta$ is a dimensionless incision-threshold number (Eq. 9). The square of this length scale is $(1 + N_\theta)\, l_c^2$, which is equal to $(h_c + \theta)/\kappa_c$, the ratio of the two intercepts of the black line. The quantity $(1 + N_\theta)\, l_c^2$ is the reciprocal of the slope of the black dashed line that connects the two intercepts. Thus, by comparing the slope of this auxiliary line and of the gray curvature–steepness-index line, we can graphically express the effect of the incision threshold on landscape dissection, as shown by the white arrow.

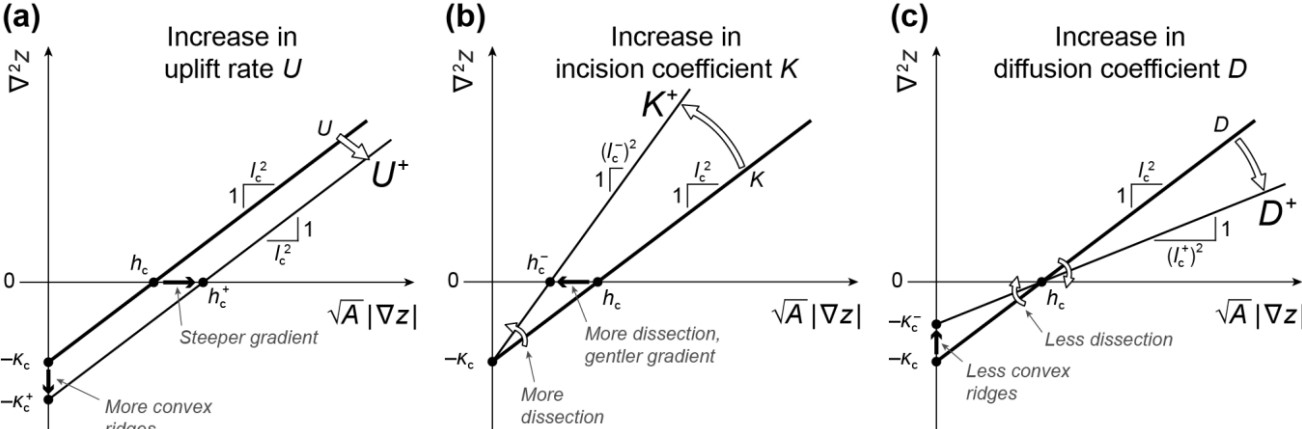

Figure 3: **Graphical illustration, using curvature–steepness-index lines, of how parameters influence landscape properties.** The three plots show how curvature–steepness index lines respond to increases in the uplift rate $U$, incision coefficient $K$, and diffusion coefficient $D$. **(a)** An increase in $U$ parallel-shifts the line to the right and downward. This makes the vertical-axis intercept smaller (more negative) and the horizontal-axis intercept bigger, showing that ridges become more convex and that gradients become steeper (i.e., relief becomes higher). The line's slope remains $1/l_c^2$, indicating that the scale of dissection does not change. **(b)** An increase of $K$ rotates the line counterclockwise around the vertical-axis intercept. This makes the horizontal-axis intercept smaller and the line's slope bigger, showing that gradients become gentler (i.e., relief becomes lower) and that the landscape becomes more dissected (i.e., the scales of dissection become smaller). **(c)** An increase of $D$ rotates the line clockwise around the horizontal-axis intercept. This moves the vertical-axis intercept closer to zero and decreases the line's slope, showing that ridges become less convex and that the landscape becomes less dissected (i.e., the scales of dissection become larger).

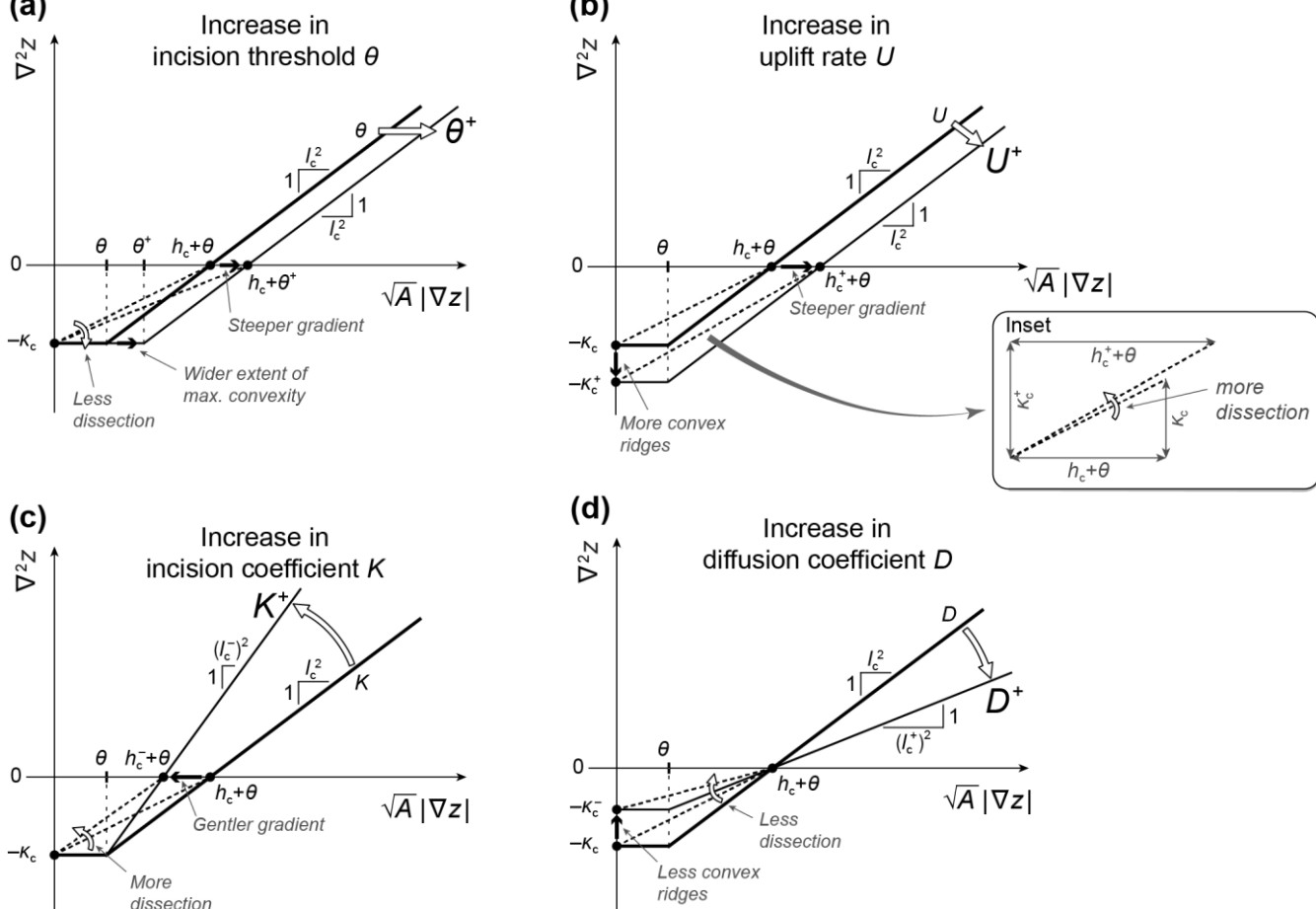

Figure 4: **Graphical illustration, using curvature–steepness-index lines, of how incision threshold and parameters control landscape properties.** The four plots show how curvature–steepness index lines respond to increases in the incision threshold $\theta$, uplift rate $U$, incision coefficient $K$, and diffusion coefficient $D$. **(a)** An increase in the incision threshold $\theta$ parallel-shifts the line to the right. Note the difference with the shift in (b), which is to the right and downward. The shift in (a) makes the horizontal segment of the line longer and the horizontal-axis intercept bigger, showing that the zones of maximum convexity become wider, gradients become steeper and drainage areas of valley heads become smaller. The curvature value of the horizontal segments and the line's slope remain unchanged. The increase in the horizontal-axis intercept changes its ratio to the vertical-axis intercept, which expresses the degree of landscape dissection as explained in Sect. 3.3. This ratio can be visualized by the dashed auxiliary lines that connect the horizontal- and vertical-axis intercepts and the change in the value of the ratio can be visualized by the rotation of these lines. **(b)** An increase in the uplift rate $U$ shifts the line to the right and downward. This makes the vertical-axis intercept smaller (more negative) and the horizontal-axis intercept bigger, showing that ridges become more convex and that gradients become steeper (i.e., relief becomes higher). The changes of these two intercepts are not proportional and, thus, their ratio changes. This change can be visualized in the inset, where we plot the auxiliary lines such that they share the same starting point. This shows that the degree of landscape dissection changes when $U$ is increased, whereas it did not change in the case of the LEM without incision threshold (see Fig. 3). **(c)** An increase in the incision coefficient $K$ rotates the inclined segment of the line counterclockwise around its intersection with the horizontal segment. The horizontal segment remains unchanged. Thus, the horizontal-axis intercept becomes smaller, which shows that gradients become gentler and the landscape becomes more dissected. **(d)** An increase in the diffusion coefficient $D$ rotates the inclined segment line clockwise around the horizontal-axis intercept. This moves the horizontal segment and the vertical-axis intercept closer to zero, and changes the ratio of the two intercepts, showing that ridges become less convex and that the landscape becomes less dissected.

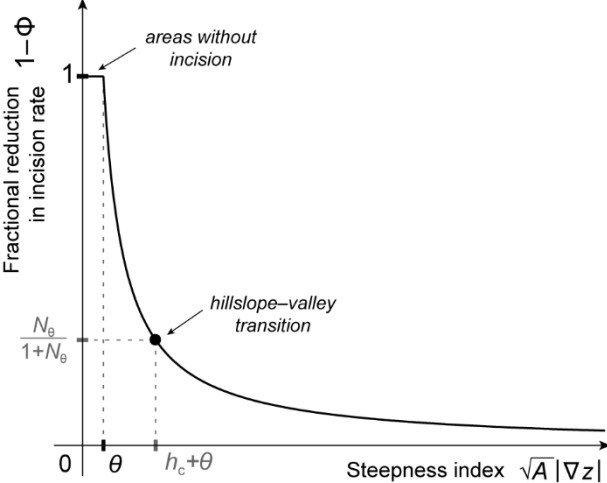

Figure 5: **How the relative influence of the incision threshold changes across a landscape.** We plot the quantity $1 - \Phi$ versus the steepness index $\sqrt{A}|\nabla z|$, where $\Phi$ is a threshold factor that can subsume the incision threshold $\theta$ (definitions in Eq. 22 and Sect. 4). The quantity $1 - \Phi$ is dimensionless and expresses the fractional reduction in incision rate due to the incision threshold $\theta$. The $1 - \Phi$ curve presented here shows how this fraction varies across the landscape. The value $1 - \Phi = 1$ corresponds to points where incision is fully suppressed by the threshold, i.e., where the incision rate is reduced by 100%. Thus, the horizontal segment of the $1 - \Phi$ curve corresponds to the horizontal segment of the curvature–steepness-index line in Fig. 2. At points with $\sqrt{A}|\nabla z|$ much larger than $\theta$ (the far right of the plot), the incision rate is reduced by a very small fraction, and the $1 - \Phi$ curve asymptotically approaches 0. The black dot on the $1 - \Phi$ curve corresponds to the steepness index value $h_c + \theta$, which corresponds to hillslope–valley transitions, as shown in Fig. 2. The position of the black dot on the curve helps us visualize how large the incision rate reduction fraction is across different regimes of a landscape. This position depends on the characteristic height $h_c$, on $\theta$, and on the dimensionless ratio $\theta/h_c$. We define this ratio as the incision-threshold number $N_\theta$ (Eq. 9).

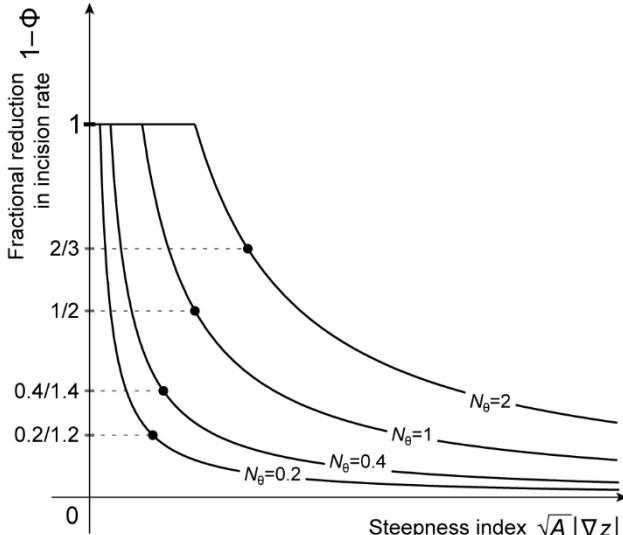

Figure 6: **Comparison of the relative influence of incision thresholds with different magnitudes.** We present curves of the quantity $1 - \Phi$ versus the steepness index $\sqrt{A}|\nabla z|$ for four different values of $N_\theta$. The black dots show the values of $\sqrt{A}|\nabla z|$ and $1 - \Phi$ that correspond to hillslope–valley transitions. The curve with $N_\theta = 0.2$, the smallest of the four values of $N_\theta$, starts with a short horizontal segment, and then descends steeply and approaches 0 rapidly. Furthermore, its black dot corresponds to the value $1 - \Phi = 0.2/1.2 = 1/6$. By contrast, the curve with the largest value, $N_\theta = 2$, starts with a long horizontal segment, descends gradually, approaches 0 slowly, and has a black dot with $1 - \Phi = 1/3 = 0.333$. These differences show that as $N_\theta$ increases, a) incision is fully suppressed by the threshold in bigger portions of hillslopes, b) the steepness index must reach greater values for the influence of the threshold to start becoming negligible; and c) the hillslope–valley transition occurs at larger values of $1 - \Phi$, i.e., the threshold has a strong influence not only on hillslopes, but also on the valley network.

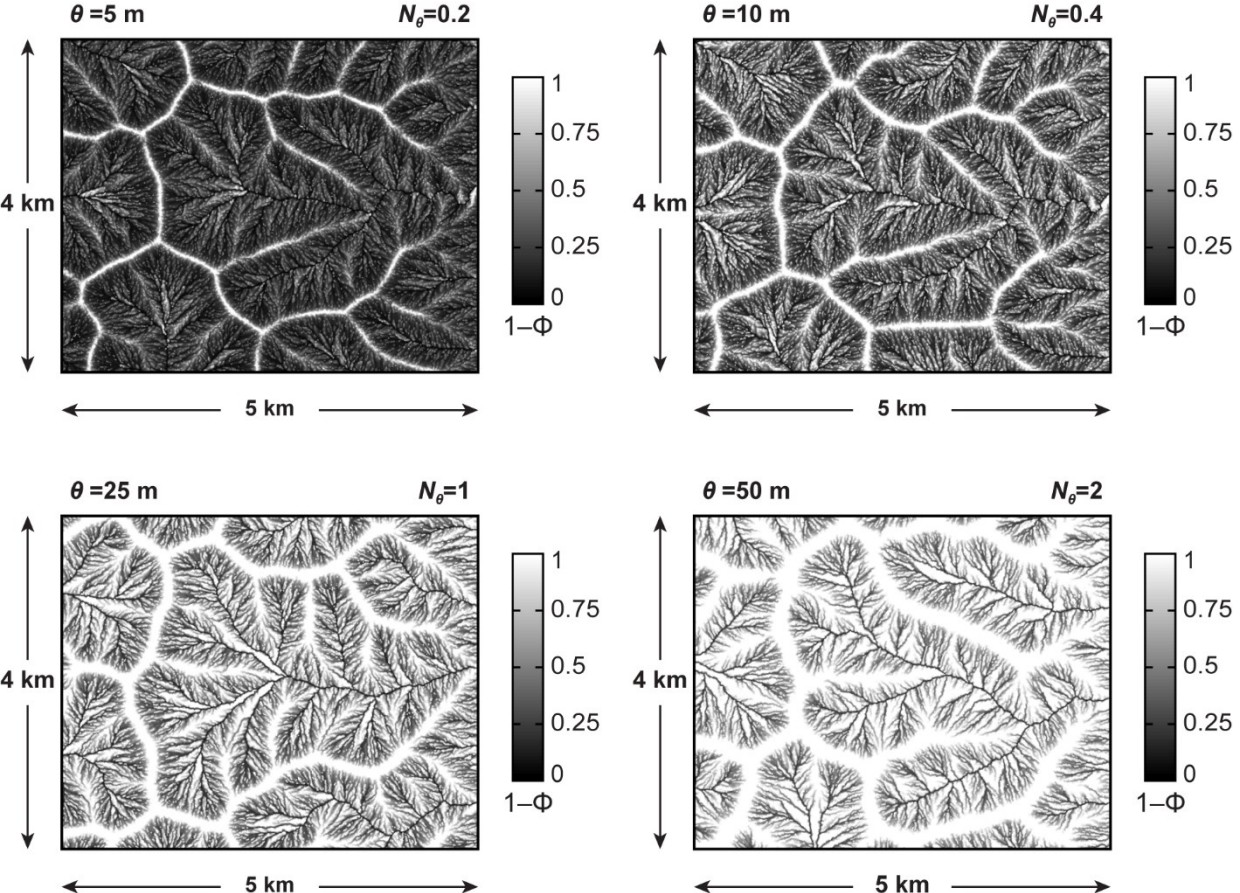

Figure 7: **Control of the incision-threshold number $N_\theta$ on the spatial distribution of the fractional reduction in incision rate.** We map the quantity $1 - \Phi$ across four steady-state simulated landscapes with different values of $N_\theta$. We use the same values of $N_\theta$ as in Fig. 6. Details about the setup and parameters of these simulations are presented in Appendix B. Lighter colors correspond to larger values of $1 - \Phi$, i.e., to stronger influence of the incision threshold. The spatial distribution of $1 - \Phi$ follows the dendritic pattern of the valley network. As $N_\theta$ increases, the maps become lighter, i.e., areas with strong influence of the incision threshold become more widespread, both on hillslopes and in valleys.