# Peer review of "Graphically interpreting how incision thresholds influence topographic and scaling properties of modeled landscapes"

_Earth Surface Dynamics, 2020_

## Referee Comment (RC1) · Fiona Clubb (Referee) · 15 Jul 2020

The paper under review expands upon previous work by the authors on performing dimensional analysis of landscape evolution models. The authors expand upon new techniques of interpreting curvature-steepness index space as a way of characterizing diffusive landscapes, similar to S-A in bedrock fluvial landscapes. They then re-define a Péclet number (competition between advection and diffusion) for landscape evolution models which take into account incision thresholds, as well as examining the influence of varying incision thresholds both between and within model domains. The paper is interesting and well-written. I found the point that, if an incision threshold is included in

the calculation of the Péclet number, the degree of landscape dissection is dependent on uplift rate very interesting and think perhaps more could be made of this in the paper. This could be a nice hypothesis to test in real landscapes where a relationship between drainage density and uplift rate has been observed.

I think the paper is suitable for publication in ESurf after a number of points (listed below) are addressed. My main issue is that more justification should be provided for the physical basis of representing the incision threshold as purely a function of area and slope, rather than as a minimum value of shear stress or stream power. It would also be good to better situate the paper in context of the wider literature, as well as clarifying the novelty of this paper compared to the authors' previous work.

Specific comments:

The introduction could better set out the novelty of the work that is being presented here. Lines 15-19 (page 2) mention the work of Theodoratos et al. (2018) and Theodoratos and Kirchner (2020), which introduce the concept of curvature-steepness space and dimensionally analyse a LEM with an incision threshold, respectively. This sounds very similar to the summary of the manuscript in the abstract, and therefore leaves the reader wondering what the novelty of this paper is compared to the previous ones.

Following on from this, it would be useful to expand the introduction with a more thorough literature review: many studies have already examined the influence of incision thresholds on erosion (e.g. Snyder et al., 2003; DiBiase and Whipple, 2011; Lague, 2014; Scherler et al., 2017; Venditti et al., 2019, etc...). This work would be better set into context with a more comprehensive review of previous studies.

Page 20, Line 30: The caveat of m=0.5 and n=1 is an important one considering that many studies have found that this is not likely to be the case in the majority of real landscapes (e.g. Lague, 2014; Harel et al. 2016). Although this caveat is mentioned here, it would be useful to expand on how changing m and n would affect the graphical interpretation of LEMs. Would it be at all possible to use curvature-steepness index

space if n is not equal to 1? Or are these tools only useful if n=1?

Page 3, Line 1, "Because a negative incision rate would not be meaningful..." doesn't a negative incision rate represent deposition? In terms of real landscapes this is meaningful.

Equation (2) sets the incision threshold $\theta$ as merely a function of area and slope, such that no incision will occur at low values of ($A\hat{}0.5*S$). What is this representing in terms of physical process? Previous approaches use incision thresholds to represent discharge variations, climatic controls on discharge, thresholds for particle motion/detachment, etc. The paper does explicitly mention this point (Page 3 Lines 5-14), and states that using this simplified formulation is more practical. However, in my opinion setting the incision threshold to be dynamic would add a lot to the paper and provide more physical basis for the parameterization.

Related to this, I found it difficult to see how the variation of the incision threshold metric (just as a function of area and slope) across the model domain would relate to the strength of incision thresholds in real landscapes. From Figure 7, it appears that this variation is just representing the distribution of area and slopes that you would expect in a landscape consisting of hillslopes and valleys. How is this related to the physical processes that would cause thresholds for incision in fluvial systems?

Page 4, Section 2.3 could be explained a bit more for readers not familiar with the previous paper. For example, how the dimensionless grouping of $K\theta/U$ was obtained.

References

DiBiase, R.A. and Whipple, K.X., 2011. The influence of erosion thresholds and runoff variability on the relationships among topography, climate, and erosion rate. Journal of Geophysical Research: Earth Surface, 116(F4).

Harel, M.A., Mudd, S.M. and Attal, M., 2016. Global analysis of the stream power law parameters based on worldwide 10Be denudation rates. Geomorphology, 268,

pp.184-196.

Lague, D., 2014. The stream power river incision model: evidence, theory and beyond. Earth Surface Processes and Landforms, 39(1), pp.38-61.

Scherler, D., DiBiase, R.A., Fisher, G.B. and Avouac, J.P., 2017. Testing monsoonal controls on bedrock river incision in the Himalaya and Eastern Tibet with a stochastic‐threshold stream power model. Journal of Geophysical Research: Earth Surface, 122(7), pp.1389-1429.

Snyder, N.P., Whipple, K.X., Tucker, G.E. and Merritts, D.J., 2003. Importance of a stochastic distribution of floods and erosion thresholds in the bedrock river incision problem. Journal of Geophysical Research: Solid Earth, 108(B2).

Venditti, J.G., Li, T., Deal, E., Dingle, E. and Church, M., 2019. Struggles with stream power: Connecting theory across scales. Geomorphology, p.106817.

---

## Referee Comment (RC2) · Philippe Steer (Referee) · 22 Sep 2020

This new paper by Theodoratos & Kirchner extends their recent developments on the dimensional analysis of landscapes at steady-state, considering here a threshold model for the stream power law. In particular, this paper offers a graphical interpretation of the findings obtained in Theodoratos & Kirchner (2020) in the form of a curvature-steepness relationship. I share most comments made by Reviewer 1 (Fiona Clubb) and do not repeat them here. Even if some redundancy exists with previous papers of the same authors, I do not find it problematic as it allows the reader to have a complete understanding of this new paper without requiring to go back and forth between these

different papers. Moreover, the paper is neat and well-written and the mathematical derivations are valid. Overall, this makes this paper readily publishable after cosmetic revisions. My review could stop here.

Despite that, I feel this is a pity that this paper only considers the dimensional analysis of modeled topographies and not natural ones. I below will try to convince the authors to make some significant additions to their paper, but the editor or the authors could judge this is not necessary.

First, my main concern is that this neat and detailed analysis is made on a model which has a weak physical basis, in particular (as fairly acknowledged by the authors) due to the use of a threshold on steepness (and not on shear stress) without a stochastic description of discharge or shear stress events. In consequence, I am left to wonder what is the real addition of a paper that considers the dimensional analysis of a model with an unsupported physical basis. Second, this paper left me wondering how the steepness-curvature analysis, two metrics that are very easily measured on DEMs, resulting from this model compares with natural landscapes. Figure 3 of Perron et al. (2019; Nature) provides a nice testable (and in my opinion promising) natural example where the theoretical-graphical predictions of the authors could apply (Figure 3 shows how curvature relates to A S - and not Aˆ0.5 S - in the Gabilan Mesa first order catchments). Using this example (or another one) to demonstrate that the curvature-steepness relationship can be used to infer some potentially new constraints on K, D and U, complementary to classical steepness analysis, would clearly represent a major addition to this paper and extend the interest of this paper to a community far wider than numerical modelers. If this works well, this would also probably help answering my first point. Indeed, this would not be the first time that a geomorphological model, with little physical support, explains well natural observations (i.e. the stream power model at steady-state with S-A relationships). However, this would give a clear support (not a physical one - but an observational one) to why we need to consider a threshold, even in its simplest form, in the stream power incision law to simulate large-scale
landscape evolution.

I sincerely hope these two comments will be perceived as encouragements and not as negative criticisms by the authors.

Philippe Steer

---

## Author Comment (AC1) · 4 Dec 2020

**Reply to referee comments on "Graphically interpreting how incision thresholds influence topographic and scaling properties of modeled landscapes" by Theodoratos and Kirchner**

We are grateful to Fiona Clubb and Philippe Steer for their feedback on our manuscript. In the following response to their reviews, we first quote their comments (in blocks of italic text) and then respond (in normal text).
Nikos Theodoratos and James Kirchner

**1. Response to referee Fiona Clubb**

> *The paper under review expands upon previous work by the authors on performing dimensional analysis of landscape evolution models. The authors expand upon new techniques of interpreting curvature-steepness index space as a way of characterizing diffusive landscapes, similar to S-A in bedrock fluvial landscapes. They then re-define a Péclet number (competition between advection and diffusion) for landscape evolution models which take into account incision thresholds, as well as examining the influence of varying incision thresholds both between and within model domains.*

Thank you for this nice summary of our work.

> *The paper is interesting and well-written.*

Thanks!

> *I found the point that, if an incision threshold is included in the calculation of the Péclet number, the degree of landscape dissection is dependent on uplift rate very interesting and think perhaps more could be made of this in the paper. This could be a nice hypothesis to test in real landscapes where a relationship between drainage density and uplift rate has been observed.*

Indeed, a relationship between drainage density and uplift rate $U$ has been observed (e.g., in your study, Clubb et al., 2016). The sign of this relationship depends on the value of the incision-term slope exponent $n$. Specifically, an increase of $U$ leads to an increase of drainage density for $n > 1$ and to a decrease of drainage density for $n < 1$ (i.e., the relationship between $U$ and drainage density is positive for $n > 1$ and negative for $n < 1$). For $n = 1$, drainage density does not depend on $U$.

We reached a similar conclusion to the above in our scaling analysis of the LEM without incision threshold in Theodoratos et al. (2018). Specifically, as described in Appendix A of that work, the characteristic length $l_c$ is proportional to $U^{(1-n)/(n+2m)}$. Thus, an increase of $U$ leads to a decrease of $l_c$ for $n > 1$ (i.e., to an increase of drainage density), to an increase of $l_c$ for $n > 1$ (i.e., to an decrease of drainage density), and to no change for $n = 1$.

To summarize, the degree of landscape dissection generally depends on the uplift rate (and, in addition, on the incision and diffusion coefficients). Dissection is independent of the uplift rate only in one special case, when there is no incision threshold and the slope exponent is $n = 1$. Interestingly, as revealed by the current study, if an incision threshold is included, the dissection depends on the uplift rate even for $n = 1$.

We will note the above points in the paper to highlight this finding.

> *I think the paper is suitable for publication in ESurf after a number of points (listed below) are addressed.*

Thank you.

> *My main issue is that more justification should be provided for the physical basis of representing the incision threshold as purely a function of area and slope, rather than as a minimum value of shear stress or stream power.*

See our response further below, under your specific comment about this issue.

> *It would also be good to better situate the paper in context of the wider literature, as well as clarifying the novelty of this paper compared to the authors' previous work.*

Having read your review, as well as the review by Philippe Steer, we realize that we have not adequately explained what were the main goals and findings of our study. The two issues raised here (the paper's novelty and position within the literature) can be clarified by revising the Introduction and Summary sections of the paper to improve the explanations of our goals and findings.

In this response, we discuss the paper's novelty and position in the literature further below, under your corresponding specific comments.

> *Specific comments:*
>
> *The introduction could better set out the novelty of the work that is being presented here. Lines 15-19 (page 2) mention the work of Theodoratos et al. (2018) and Theodoratos and Kirchner (2020), which introduce the concept of curvature-steepness space and dimensionally analyse a LEM with an incision threshold, respectively. This sounds very similar to the summary of the manuscript in the abstract, and therefore leaves the reader wondering what the novelty of this paper is compared to the previous ones.*
>
> *Following on from this, it would be useful to expand the introduction with a more thorough literature review: many studies have already examined the influence of incision thresholds on erosion (e.g. Snyder et al., 2003; DiBiase and Whipple, 2011; Lague, 2014; Scherler et al., 2017; Venditti et al., 2019, etc...). This work would be better set into context with a more comprehensive review of previous studies.*

The main goal of our paper was to explore the explanatory power of the relationship between curvature and the steepness index, specifically, the visual explanatory power of plots of this relationship. We found that simple shifts and rotations of these plots express graphically how scaling properties of landscapes respond to changes in the values of parameters of the LEM. The scaling properties that we examine are quantified by the characteristic scales of length and height.

To add more texture to this graphical method, we also examined the LEM with incision threshold. We found that changes in the value of the incision threshold, too, lead to changes in scaling properties that are graphically expressed as shifts and rotations.

In Theodoratos et al. (2018), we showed how the characteristic scales of length and height can be used to non-dimensionalize the LEM without incision threshold. In Theodoratos and Kirchner (2020), we showed that the same characteristic scales can be used to non-dimensionalize the LEM that includes an incision threshold. Thus, these two studies laid the ground for the current work. Furthermore, we used the curvature–steepness-index relationship in Theodoratos et al. (2018) to derive geomorphologic interpretations of the characteristic scales. However, we used this relationship to visualize the response of landscape properties in the current work for the first time.

Additionally, in Theodoratos and Kirchner (2020) we defined the dimensionless incision-threshold number $N_\theta$ and we showed that it can be used to compare the relative importance of the incision threshold across different landscapes. On the other hand, in the current work, we defined the dimensionless quantity $1 - \Phi$ (the fractional reduction in incision rate), which quantifies how the relative influence of the incision threshold varies within a given landscape.

The current work has greatly benefited from studies of the influence of the incision threshold on erosion, such as the ones that you mention. However, our work focuses primarily on the incision threshold's influence on scaling properties, and on how this influence can be graphically visualized. Therefore, the position of our paper is at the intersection of the literature on incision thresholds, on landscape scaling, and on graphical methods.

As mentioned above, we realize now that we did not succeed in explaining our work's novelty and position in the literature, and so we will improve the Introduction and Summary.

> *Page 20, Line 30: The caveat of m=0.5 and n=1 is an important one considering that many studies have found that this is not likely to be the case in the majority of real landscapes (e.g. Lague, 2014; Harel et al. 2016). Although this caveat is mentioned here, it would be useful to expand on how changing m and n would affect the graphical interpretation of LEMs. Would it be at all possible to use curvature-steepness index space if n is not equal to 1? Or are these tools only useful if n=1?*

This is a very good question, thank you.

Our graphical methods work for any value of the exponents *m* and *n*. We will add an appendix to the paper to demonstrate this. Here, we briefly present the basic equations.

For generic drainage area and slope exponents *m* and *n*, the governing equation is:

$$\frac{\partial z}{\partial t} = \begin{cases} D\nabla^2 z + U \ , & A^m(|\nabla z|)^n \le \theta \\ -K(A^m(|\nabla z|)^n - \theta) + D\nabla^2 z + U \ , & A^m(|\nabla z|)^n > \theta \end{cases} .$$

Given that the steepness index is defined as $k_s = A^{m/n}|\nabla z|$, the quantity $A^m(|\nabla z|)^n$ in the above equation is equal to the steepness index raised to the power *n*.

Setting $\partial z/\partial t = 0$ in this governing equation, we can derive the steady-state relationship between curvature and the steepness index:

$$\begin{cases} \nabla^2 z = -\dfrac{U}{D} \ , & A^m(|\nabla z|)^n \le \theta \\ \nabla^2 z = \dfrac{K}{D} A^m(|\nabla z|)^n - (1 + N_\theta)\dfrac{U}{D} \ , & A^m(|\nabla z|)^n > \theta \end{cases} ,$$

where $N_\theta$ is the incision-threshold number, defined as $N_\theta = K\theta/U$.

This relationship has the same basic properties as the relationship presented in our manuscript. It plots as a line with two segments, a horizontal segment for $A^m(|\nabla z|)^n \leq \theta$ and an inclined segment for $A^m(|\nabla z|)^n > \theta$ with slope equal to $K/D$ and intercept equal to $(U/K) + \theta$. This line responds to changes in the values of parameters with shifts and rotations, as shown in Fig. 4 of our manuscript, and these shifts and rotations express changes in the scaling properties of the landscape.

Note that, in the case of generic exponents $m$ and $n$, the characteristic scales of length and height are not equal to the parameter ratios $K/D$ and $U/K$, rather they are defined by more complicated formulas, which can be seen in Appendix A of Theodoratos et al. (2018). However, the ratios $K/D$ and $U/K$ still express the relative strengths of incision, diffusion, and uplift.

Finally, for generic exponents $m$ and $n$, the fractional reduction in incision rate $1 - \Phi$ is defined as

$$1 - \Phi = \begin{cases} 1 & , \quad A^m(|\nabla z|)^n \leq \theta \\ \dfrac{\theta}{A^m(|\nabla z|)^n} & , \quad A^m(|\nabla z|)^n > \theta \end{cases},$$

which plots as shown in Figs. 5 and 6, but in axes of $A^m(|\nabla z|)^n$, i.e., of steepness index raised to the power $n$.

> *Page 3, Line 1, "Because a negative incision rate would not be meaningful..." doesn't a negative incision rate represent deposition? In terms of real landscapes this is meaningful.*

Indeed, this would be meaningful in a landscape that is influenced by deposition. However, our model is based on the assumption of detachment-limited sediment transport, where deposition does not occur. We will add a clarification about this in the paper.

> *Equation (2) sets the incision threshold θ as merely a function of area and slope, such that no incision will occur at low values of (A^0.5*S). What is this representing in terms of physical process? Previous approaches use incision thresholds to represent discharge variations, climatic controls on discharge, thresholds for particle motion/detachment, etc. The paper does explicitly mention this point (Page 3 Lines 5-14), and states that using this simplified formulation is more practical. However, in my opinion setting the incision threshold to be dynamic would add a lot to the paper and provide more physical basis for the parameterization.*

Physically, our formulation represents a landscape where precipitation is uniform in space and constant in time. In such a case, any given combination of drainage area $A$ and slope $|\nabla z|$ would lead to the same value of stream power (or shear stress) for any storm event (as all events would be equal), and this value of stream power would either be above or below the incision threshold. Thus, in such an idealized case, defining a topographic threshold based on $\sqrt{A}|\nabla z|$ would be exactly equivalent to defining a threshold of stream power (or shear stress).

This idealized formulation represents a first-order approximation of the time-averaging that is inherent in mathematically describing how landscapes evolve in geomorphic time scales under the influence of processes that operate in hydrologic time scales.

> *Related to this, I found it difficult to see how the variation of the incision threshold metric (just as a function of area and slope) across the model domain would relate to the*

*strength of incision thresholds in real landscapes. From Figure 7, it appears that this variation is just representing the distribution of area and slopes that you would expect in a landscape consisting of hillslopes and valleys. How is this related to the physical processes that would cause thresholds for incision in fluvial systems?*

Indeed, the spatial patterns in Fig. 7 reflect the distribution of area and slopes, because the incision threshold is defined as a topographic threshold, as described in our previous comment. However, the metric $1 - \Phi$ (fractional reduction in incision rate) has a more general usefulness. For instance, Tucker (2004) examined an incision threshold formulation that assumed stochastic precipitation, and introduced the threshold factor $\Phi$ (on which the notation of our metric $1 - \Phi$ was based). Using Tucker's formulation of the incision term and definition of $\Phi$, one could still draw a map of $1 - \Phi$ to visualize the spatial distribution of the relative influence of the incision threshold.

We will add a note in the paper to clarify this.

*Page 4, Section 2.3 could be explained a bit more for readers not familiar with the previous paper. For example, how the dimensionless grouping of $K\theta/U$ was obtained.*

This grouping emerged out of the non-dimensionalization of the governing equation with incision threshold. We mention this in Sect. 2.3, but we will improve our wording to make this clearer.

**2. Response to referee Philippe Steer**

*This new paper by Theodoratos & Kirchner extends their recent developments on the dimensional analysis of landscapes at steady-state, considering here a threshold model for the stream power law. In particular, this paper offers a graphical interpretation of the findings obtained in Theodoratos & Kirchner (2020) in the form of a curvature-- steepness relationship. I share most comments made by Reviewer 1 (Fiona Clubb) and do not repeat them here. Even if some redundancy exists with previous papers of the same authors, I do not find it problematic as it allows the reader to have a complete understanding of this new paper without requiring to go back and forth between these different papers. Moreover, the paper is neat and well-written and the mathematical derivations are valid. Overall, this makes this paper readily publishable after cosmetic revisions. My review could stop here.*

Thank you for the supportive comments!

*Despite that, I feel this is a pity that this paper only considers the dimensional analysis of modeled topographies and not natural ones. I below will try to convince the authors to make some significant additions to their paper, but the editor or the authors could judge this is not necessary.*

We agree that an application of our method to natural topographies would be very beneficial, but this would require significant additions to the paper as you write. We believe that the analysis of natural landscapes should be pursued as a separate study. This way, the results based on theory and those based on natural landscapes would not obscure each other.

We thank you for trying to convince us to include an analysis of real landscapes. This is a very reasonable next step, and we realize that our manuscript is not making this sufficiently clear.

Therefore, we will highlight this point in Sect. 3, where the curvature–steepness-index relationship is defined, and in the summary of the manuscript.

> *First, my main concern is that this neat and detailed analysis is made on a model which has a weak physical basis, in particular (as fairly acknowledged by the authors) due to the use of a threshold on steepness (and not on shear stress) without a stochastic description of discharge or shear stress events. In consequence, I am left to wonder what is the real addition of a paper that considers the dimensional analysis of a model with an unsupported physical basis.*

As we wrote above in response to referee Fiona Clubb, the main goal of this paper was to demonstrate the graphical explanatory power of the curvature–steepness-index relationship for landscapes that are influenced by diffusion. We argue that this relationship can be viewed as a counterpart to the slope–area power-law relationship, which is routinely used to analyze landscapes that are shaped by incision and uplift, but not landscapes that include diffusion.

We recognize that an LEM with a stochastic incision threshold would follow a different curvature–steepness-index relationship than the one described in our manuscript. However, the system of axes of curvature and the steepness index would still be a useful space to plot data in, provided that the LEM would also include diffusion. Indeed, this system of axes leads to meaningful plots for both the LEM that includes the non-stochastic formulation of the incision threshold and the LEM that does not include an incision threshold at all.

Based on your comment, however, we realize that our paper can become more convincing if we clarify this issue. So, we thank you for this comment!

> *Second, this paper left me wondering how the steepness-curvature analysis, two metrics that are very easily measured on DEMs, resulting from this model compares with natural landscapes. Figure 3 of Perron et al. (2019; Nature) provides a nice testable (and in my opinion promising) natural example where the theoretical-graphical predictions of the authors could apply (Figure 3 shows how curvature relates to A S - and not A^0.5 S - in the Gabilan Mesa first order catchments). Using this example (or another one) to demonstrate that the curvature--steepness relationship can be used to infer some potentially new constraints on K, D and U, complementary to classical steepness analysis, would clearly represent a major addition to this paper and extend the interest of this paper to a community far wider than numerical modelers. If this works well, this would also probably help answering my first point. Indeed, this would not be the first time that a geomorphological model, with little physical support, explains well natural observations (i.e. the stream power model at steady-state with S-A relationships). However, this would give a clear support (not a physical one - but an observational one) to why we need to consider a threshold, even in its simplest form, in the stream power incision law to simulate large-scale landscape evolution.*

As we wrote above, our study aims primarily at the usefulness of the curvature–steepness-index relationship, not at the incision threshold. Furthermore, we believe that the aims of an analysis of real landscapes can be best pursued in a separate manuscript. However, this is a very reasonable suggestion, and we need to highlight it more in our manuscript.

As an aside, we happily note that you appear to have read our work and the related literature with care, since you have observed that Figure 3 in Perron et al. (2009) shows a relationship of curvature versus $A|\nabla z|$ not versus $\sqrt{A}|\nabla z|$. Our plots of curvature versus $\sqrt{A}|\nabla z|$ are more closely related to Perron et al.'s (2009) Figure 3 (b), a plot of the so-called slope function $S^*$ versus drainage area $A$, which is defined by Perron et al.'s (2009) Eq. (5), which is a rearranged version of our curvature–steepness-index relationship.

> *I sincerely hope these two comments will be perceived as encouragements and not as negative criticisms by the authors.*

Your comments are reasonable and very helpful. Therefore, they are very welcome!

**References**

Clubb, F. J., Mudd, S. M., Attal, M., Milodowski, D. T., and Grieve, S. W. D.: The relationship between drainage density, erosion rate, and hilltop curvature: Implications for sediment transport processes, J. Geophys. Res. Earth Surf., 121(10), 1724–1745, doi:10.1002/2015JF003747, 2016.

Perron, J. T., Kirchner, J. W., and Dietrich, W. E.: Formation of evenly spaced ridges and valleys, Nature, 460(7254), 502–505, doi:10.1038/nature08174, 2009.

Theodoratos, N. and Kirchner, J. W.: Dimensional analysis of a landscape evolution model with incision threshold, Earth Surf. Dynam., 8, 505–526, https://doi.org/10.5194/esurf-8-505-2020, 2020.

Theodoratos, N., Seybold, H. and Kirchner, J. W.: Scaling and similarity of a stream-power incision and linear diffusion landscape evolution model, Earth Surf. Dyn., 6(3), 779–808, https://doi.org/10.5194/esurf-6-779-2018, 2018.

Tucker, G. E.: Drainage basin sensitivity to tectonic and climatic forcing: Implications of a stochastic model for the role of entrainment and erosion thresholds, Earth Surf. Proc. Landforms, 29(2), 185–205, DOI: 10.1002/esp.1020, 2004.

---

## Referee Report (RR1)

**Re-review of "Graphically interpreting how incision thresholds influence topographic and scaling properties of modeled landscapes" by Theodoratos and Kirchner, ESurfD, 2020**

The authors have done a very good job of addressing the comments I outlined in the first review, including outlining the novelty of their work in the introduction and adding an appendix that explores the generic case of m and n ≠ 1. The use of the area and slope incision threshold is more clearly explained and set in better context with other studies which use stochastic precipitation thresholds.

I still think there could be more acknowledgement of the literature on incision thresholds and exploration of other studies which have explored the topographic signatures of incision versus diffusion, which are mentioned very briefly in the introduction. However, I see the point that this paper is combining a few different approaches and might end up becoming very long, so I won't push this further.

I also agree with Philippe Steer's comments from the first round of revision that applying this to real landscapes would have been of significant benefit to the paper, although acknowledge that it would have significantly lengthened the manuscript.

I recommend the paper be published in its current form and look forward to seeing the final version in ESurf!